# Cryptic terrestrial fungus-like fossils of the early Ediacaran Period

Tian Gan[1,2,3,4], Taiyi Luo[1✉], Ke Pang[3,4✉], Chuanming Zhou[3], Guanghong Zhou[5], Bin Wan[3], Gang Li[6], Qiru Yi[4], Andrew D. Czaja[7] & Shuhai Xiao[2✉]

The colonization of land by fungi had a significant impact on the terrestrial ecosystem and biogeochemical cycles on Earth surface systems. Although fungi may have diverged ~1500–900 million years ago (Ma) or even as early as 2400 Ma, it is uncertain when fungi first colonized the land. Here we report pyritized fungus-like microfossils preserved in the basal Ediacaran Doushantuo Formation (~635 Ma) in South China. These micro-organisms colonized and were preserved in cryptic karstic cavities formed via meteoric water dissolution related to deglacial isostatic rebound after the terminal Cryogenian snowball Earth event. They are interpreted as eukaryotes and probable fungi, thus providing direct fossil evidence for the colonization of land by fungi and offering a key constraint on fungal terrestrialization.

[1] State Key Laboratory of Ore Deposit Geochemistry, Institute of Geochemistry, Chinese Academy of Sciences, Guiyang, China. [2] Department of Geosciences, Virginia Tech, Blacksburg, VA, USA. [3] State Key Laboratory of Palaeobiology and Stratigraphy, Nanjing Institute of Geology and Palaeontology and Center for Excellence in Life and Paleoenvironment, Chinese Academy of Sciences, Nanjing, China. [4] University of Chinese Academy of Sciences, Beijing, China. [5] School of Geography and Resources, Guizhou Education University, Guiyang, China. [6] Institute of High Energy Physics, CAS, Beijing, China. [7] Department of Geology, University of Cincinnati, Cincinnati, OH, USA. ✉email: luotaiyi@mail.gyig.ac.cn; kepang@nigpas.ac.cn; xiao@vt.edu

The terrestrialization of planet Earth is a critical transition in the history of life. Fungi played multi-faceted roles in this transition[1], as they had a strong influence on continental weathering[2,3], global biogeochemical cycles[1], and ecological interactions among terrestrial organisms[3]. Constraining the timeline of fungal terrestrialization is crucial to understand the development of terrestrial ecosystems. Molecular clock studies indicate fungi diverged at ~1500–900 Ma[4–6]. However, the pre-Devonian fossil record of fungi is scarce[7,8]. Some examples include putative fungal fossils from the ~2.4 Ga Ongeluk Formation in South Africa[9], 1010–890 Ma Grassy Bay Formation in Canada[10], 810–715 Ma Mbuji-Mayi Supergroup in Democratic Republic of Congo[11], ~850 Ma Wynniatt Formation in Canada[12], and the ~600 Ma Doushantuo Formation in South China[13]. However, these fossils came from marine and estuarine environments and they have not been unambiguously shown to be terrestrial fungi[8], although some of them were allegedly transported from terrestrial environments[10]. The Silurian–Devonian organic-walled microfossil *Tortotubus protuberans* may represent a dikaryotic fungus in marginal marine or terrestrial (fluvial or lacustrine) environments[14], but undisputed terrestrial fungal fossils first appeared in the early Devonian (410 Ma) Rhynie chert[15]. The Rhynie chert contains a phylogenetically diverse fungal assemblage that includes representatives of the Chytridiomycota, Blastocladiomycota, Glomeromycotina, and Ascomycota[7,15]. The diversity of Silurian–Devonian terrestrial fungi implies the existence of fungi in pre-Silurian terrestrial ecosystems, a prediction consistent with molecular clock estimates[5] but not yet conclusively confirmed by the fossil record.

Here we report pyritized fungus-like microfossils preserved in silica cements that fill sheet-cavities (or largely bedding parallel fissures) in the cap dolostone of the basal Ediacaran Doushantuo Formation at Weng'an, South China (Fig. 1; Supplementary Fig. 1). The age of cap dolostone deposition is tightly bracketed to be ca. 635 Ma by U-Pb zircon ages[16,17]. The sheet-cavities may have been initiated physically as sheet-cracks[18,19], which were subsequently augmented by karstic dissolution related to post-deglacial isostatic rebound after the terminal Cryogenian snowball Earth event[20–22]. They are filled with at least three generations of cement: isopachous dolomite cement followed by microsparic calcite cement (which is subsequently replaced by hydrothermal chalcedony[22]) and blocky calcite cement. These cements were inferred to have formed shortly after the paleo-karstic dissolution[21] and before the deposition of ~632 Ma strata overlying the cap dolostone (Supplementary Notes 1–3). A detailed description of the Doushantuo cap dolostone, age constraints, sheet-cavity cements is provided in Supplementary Notes 1–3, and a model for sheet-cavity formation and cementation is shown in Supplementary Figure 2 and Supplementary Note 4.

## Results

The microfossils are pyritized but contain trace amount of organic matter. They occur mostly in chalcedony cement within the sheet-cavities at the Datang and Beidoushan sections in the Weng'an area (Fig. 1). They were observed and characterized in thin sections using a combination of in situ analytical techniques, including transmitted light microscopy (TLM), synchrotron radiation X-ray tomographic microscopy (SRXTM), confocal laser scanning microscopy (CLSM), focused ion beam scanning electron microscopy (FIB-SEM), energy dispersive X-ray spectroscopy (EDS), secondary ion mass spectrometry (SIMS), Fourier transform-infrared spectroscopy (FTIR), Raman spectroscopy, and other research tools (see Methods).

The microfossils include branching filaments (Figs. 2; 3a–e; 4b, g–k; Supplementary Fig. 3a–f, j) and associated hollow spheres (Figs. 2a, c, g, k; 3f, g; 4; Supplementary Fig. 3a, g–i). The filaments can be described under two morphological types, Type A and Type B, based on their difference in thickness. Type A filaments (Figs. 2; 4b, g–k; 5a; Supplementary Fig. 3a–f, j) are thicker, 5.0–9.0 μm in diameter (average = 6.8 μm; SD = 1.0 μm; $n = 119$; Fig. 5c), with associated small spheres 15.4–25.7 μm in diameter (average = 19.6 μm; SD = 2.9 μm; $n = 13$; Figs. 2a, c, g; 4a–c, h–k; Supplementary Fig. 3a, g–i) and relatively large spheres 36.0–101.7 μm in diameter (average = 63.1 μm; SD = 11.1 μm; $n = 58$; Figs. 2k; 4d–h). Type B filaments (Figs. 3a–e; 5b) are thinner, 2.0–3.4 μm in diameter (average = 2.7 μm; SD = 0.4 μm; $n = 80$; Fig. 5c), and associated with small spheres 10.3–19.9 μm in diameter (average = 16.5 μm; SD = 3.1 μm; $n = 7$; Fig. 3f, g). Filaments of both types can be straight, curved, or bent. They are hundreds of microns in length at the minimum; full length was not measured because thin sections only captured a segment of the filaments. Although most filaments are completely pyritized and are thus opaque under TLM (Figs. 2a, c, g–k; 3a–e; 4c–k; Supplementary Fig. 3e, h, j), some are partly pyritized and are thus translucent (Figs. 2b, d–f; 3f, g; 4a, b; Supplementary Fig. 3a–d, f, g, i). It is clear in these translucent specimens that the filaments are not septate. An axial strand up to half of the full filament diameter may be present in some translucent Type A filaments (Fig. 2c–e).

Filaments of both types branch frequently (Figs. 2a–h; 3a–e; 4b, g–i, k; Supplementary Fig. 3a–f, j), with acute (Figs. 2a–c; 3a–c; 4b, g–i; Supplementary Fig. 3a, b, d, j) to nearly orthogonal branching angles (Figs. 2b, d–i, 3c–e; 4k; Supplementary Fig. 3b, c). Branching style can be dichotomous (Figs. 2a; 3a; 4g–i) or monopodial (Figs. 2b, d, e; 3b, c; Supplementary Fig. 3b, c). Some monopodially branching specimens have a short lateral branch that can be either straight (Fig. 2d, e, arrows; Supplementary Fig. 3b, arrow) or curved (Fig. 2b, arrow; Supplementary Fig. 3c, arrow). Multiple orders of branches are present in some specimens (Figs. 2a–e, g, h; 3a, b). There is clear evidence for fusion, with some short lateral branches approaching toward each other (Fig. 2e, arrows) whereas others are completely fused to form an A-, H-, or ladder-like branching system (Figs. 2f–i; 3c–e; Supplementary Fig. 3d). In many cases, extensive branching and fusion of filaments resulted in mycelium-like networks (Fig. 2j, k; Supplementary Fig. 3e, f).

The spheres are hollow (Figs. 3f; 4a–c, e, g, j; Supplementary Fig. 3h, i). The smaller spheres are coaxially aligned with and physically attached to a filament, either intercalarily (Figs. 2g; 3f; 4a, b, i–k; Supplementary Fig. 3g, i), or terminally (Figs. 2c; 3g; 4f), or in a string (Fig. 4c; Supplementary Fig. 3h); there appears to be a septum between the filament and the coaxially attached sphere (Figs. 3f; 4a, j), although this cannot be unambiguously confirmed. Most of the larger spheres are in tangential contact with one or more filaments (Figs. 2k; 4g, h), although there are rare examples where the filaments clearly penetrate through the coaxially attached spheres (Fig. 4e, g, h).

The microfossils appear as translucent to opaque structures embedded in transparent chalcedony or $SiO_2$ (Supplementary Fig. 4), and they are sometimes cut by chalcedony botryoids (Fig. 2a; Supplementary Figs. 3a; 4j) and late diagenetic veins (Supplementary Fig. 3j). They are three-dimensionally pyritized, as revealed by EDS elemental maps of FIB-manufactured and petrographic sections (Supplementary Figs. 5; 6), but some contain a trace amount of organic matter, as documented by FTIR data (Fig. 6) and Raman spectroscopic data (Fig. 7). Fossil-replicating pyrite is present as translucent nanocrystallites (~500 nm) or opaque euhedral-subhedral crystals (1–5 μm). The $\delta^{34}S_{V-CDT}$ values of pyrite from Type A filaments are 9.2‰–18.7‰ (average = 14.6‰; SD = 2.6‰; $n = 21$; Supplementary Fig. 7; Supplementary Table 1).

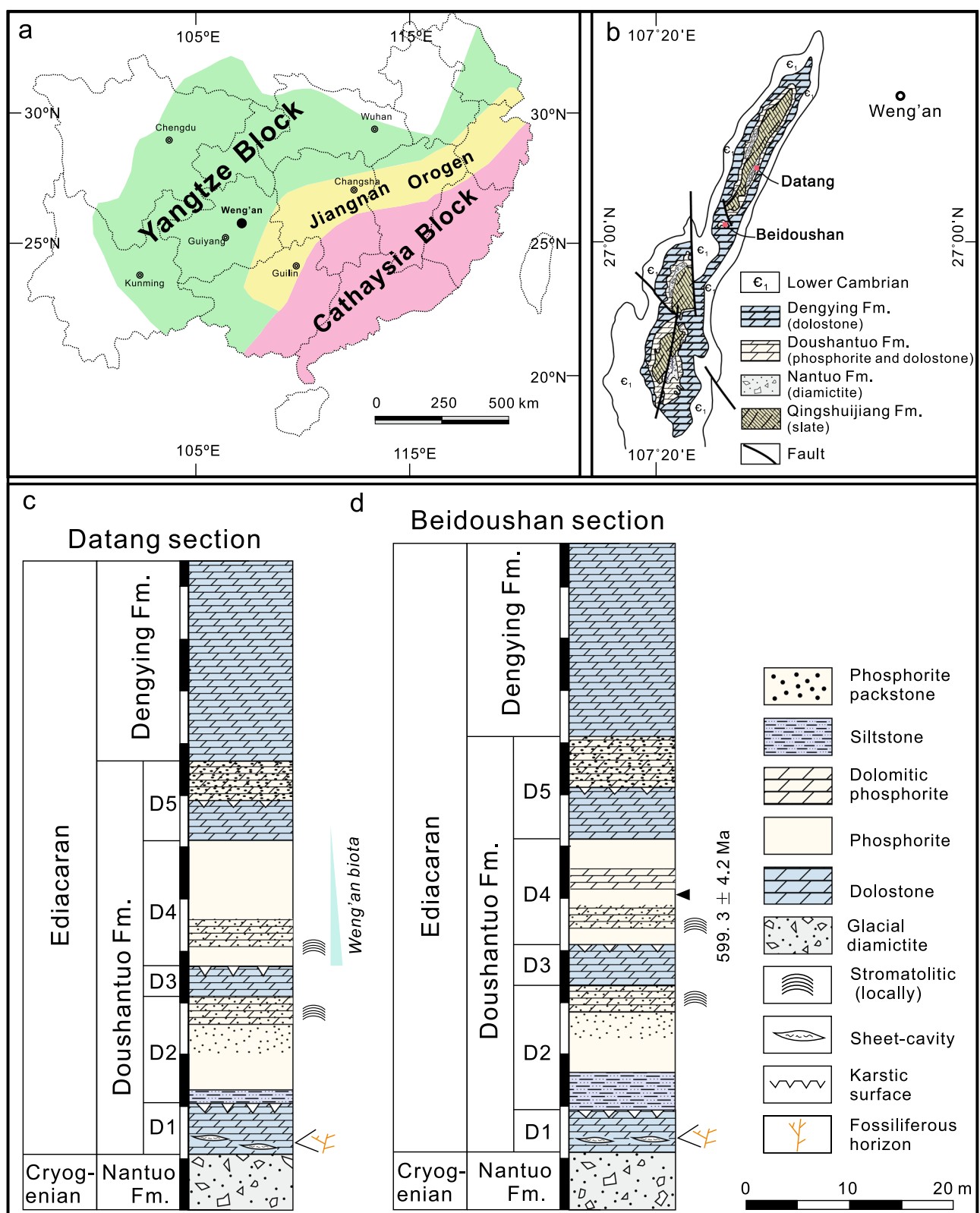

**Fig. 1 Geological maps, sample locality, and stratigraphic columns of terminal Cryogenian Nantuo Formation and Ediacaran Doushantuo-Dengying formations at Datang and Beidoushan sections, Weng'an, Guizhou Province, South China. a** Map showing major tectonic units and location of Weng'an area in South China, drawn by authors. **b** Geological map of Weng'an area, adopted from ref. [78] with permission, showing sample locality at Datang and Beidoushan. **c**, **d** Stratigraphic columns of Datang (**c**) and Beidoushan (**d**) sections. Stratocolumn of Datang section drawn by authors based on the description in ref. [79] and stratocolumn of Beidoushan section adopted from ref. [13] with permission. Unit D1 is the cap dolostone. Fm. = Formation. Data source of radiometric age: 599.3 ± 4.2 Ma—ref. [80]. See Supplementary Note 1 for a detailed stratigraphic description.

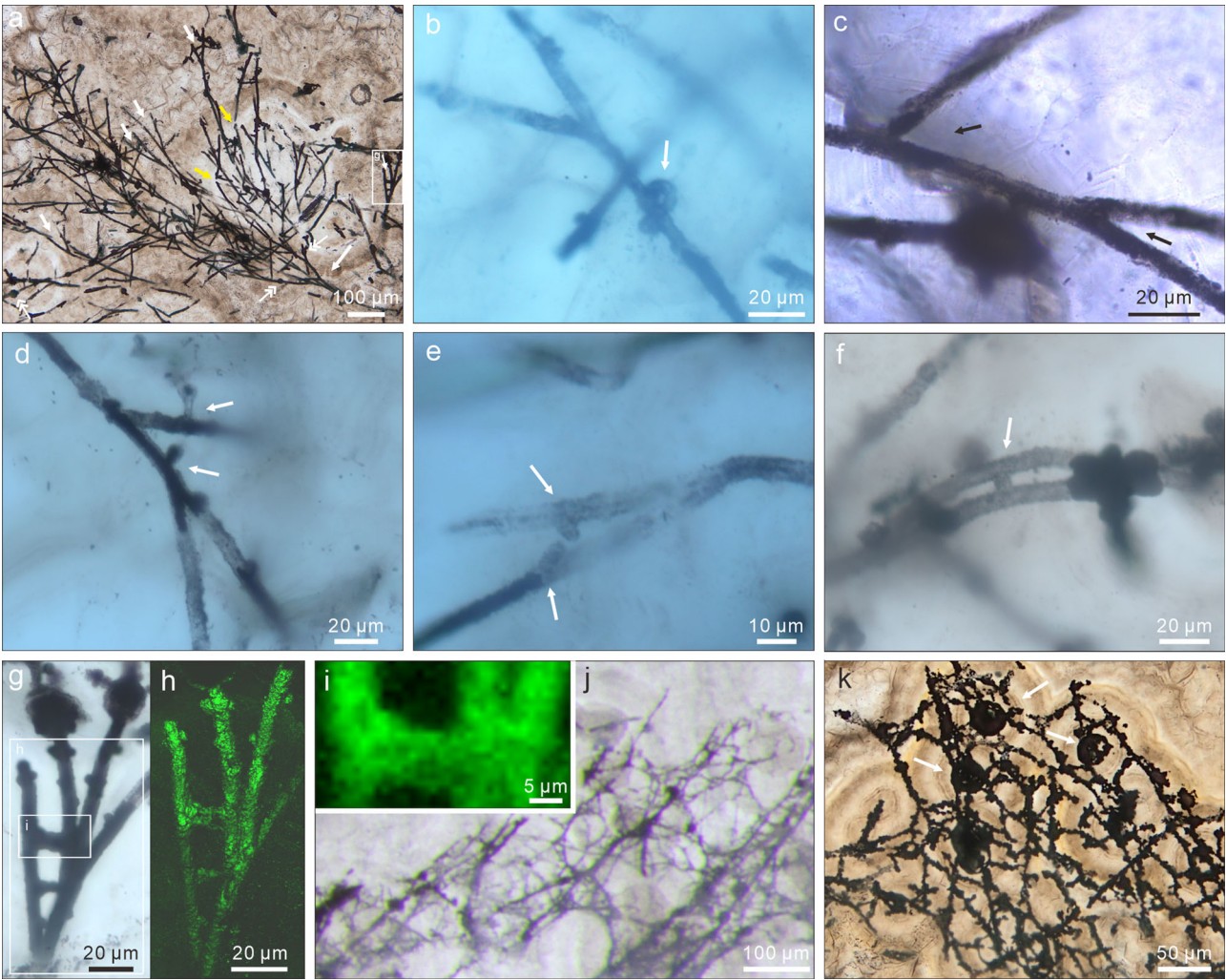

**Fig. 2 TLM, CLSM, and Raman micrographs of Type A filaments and associated spheres. a** Aggregate of Type A filaments associated with small spheres. Filaments are embedded in and sometimes cut by chalcedony botryoids (yellow arrows). Note branching filaments (white arrows), ladder-like branching systems (uppermost and rightmost white arrows), and small spheres (double-headed white arrows). **b–d** Filaments with multiple orders of branching (e.g., arrows in **c**). Note short lateral branches (arrows in **b** and **d**) and small sphere (lower central in **c**). **e** Branching filaments with two short, secondary lateral branches (arrows) approaching toward each other. **f** A-like branching system (arrow). **g** Magnification of central right in **a**, showing ladder-like branching system and two small spheres coaxially aligned with filaments. **h** CLSM micrograph corresponding to larger box in **g** (see Supplementary Movie 1). **i** Raman map of pyrite (peak at ~380 cm$^{-1}$), corresponding to smaller box in **g**. **j**, **k** Anastomosed networks of filaments. Arrows in **k** denote associated larger spheres. For each illustrated specimen in this and other figures, its repository information is given in Supplementary Table 3.

## Discussion

The microfossils exhibit morphological features that rule out abiotic origins. Filamentous structures with consistent morphologies, including multiple orders of branches, curved filaments, A-, H-, and ladder-like branching systems, and filament networks, are inconsistent with abiotic origins. Some inorganic mineral filaments may exhibit a filamentous morphology[23–25], but they seldom branch. Fractal mineral "biomorphs" do not form consistent A-, H-, or ladder-like branches, or filamentous networks[25]. The gently curved (Figs. 2a, f; 3a, d, g; 4g–i) and strongly bent filaments (Fig. 2b; Supplementary Fig. 3c) are consistent with flexible organic filaments[26], rather than rigid mineral filaments. A previous study shows that the size distributions of mineral "biomorphs" are unimodal and wide, with an average/standard deviation ratio (A/SD) of 1.7–2.7, whereas the size distributions of microbes are comparatively narrower (A/SD = 4.9), unimodal for single-species populations, and plurimodal for multi-species assemblages[25]. For comparison, the

diameter of the Doushantuo filaments has a bimodal distribution (Fig. 5c), with A/SDs of 7.0 and 7.5 for Type A and Type B, respectively, suggesting that they represent two species of microorganisms.

Several groups of fossil and extant micro-organisms provide potential interpretive analogs for the Doushantuo filaments described here. Pyritized filamentous fossils from the ~3235 Ma Sulphur Springs Group[27] and the ~1800 Ma Duck Creek Formation[28] in Australia, the latter interpreted as sulfur bacteria, are similar to the Doushantuo filaments in terms of filament size and preservation, but these fossil filaments do not branch or fuse; similarly, extant filamentous sulfur bacteria (for example, *Beggiatoa* and *Thioploca*[29]) do not develop any branches either[29]. Filamentous microfossils preserved in hydrothermal quartz veins cutting karstic dolostones of the late Ediacaran Qigebulake Formation in northwestern Tarim Basin were interpreted as Fe-oxidizing bacteria living in hydrothermal vents[30]. These filaments are encrusted by Fe-(oxyhydr)oxide, which may be weathering

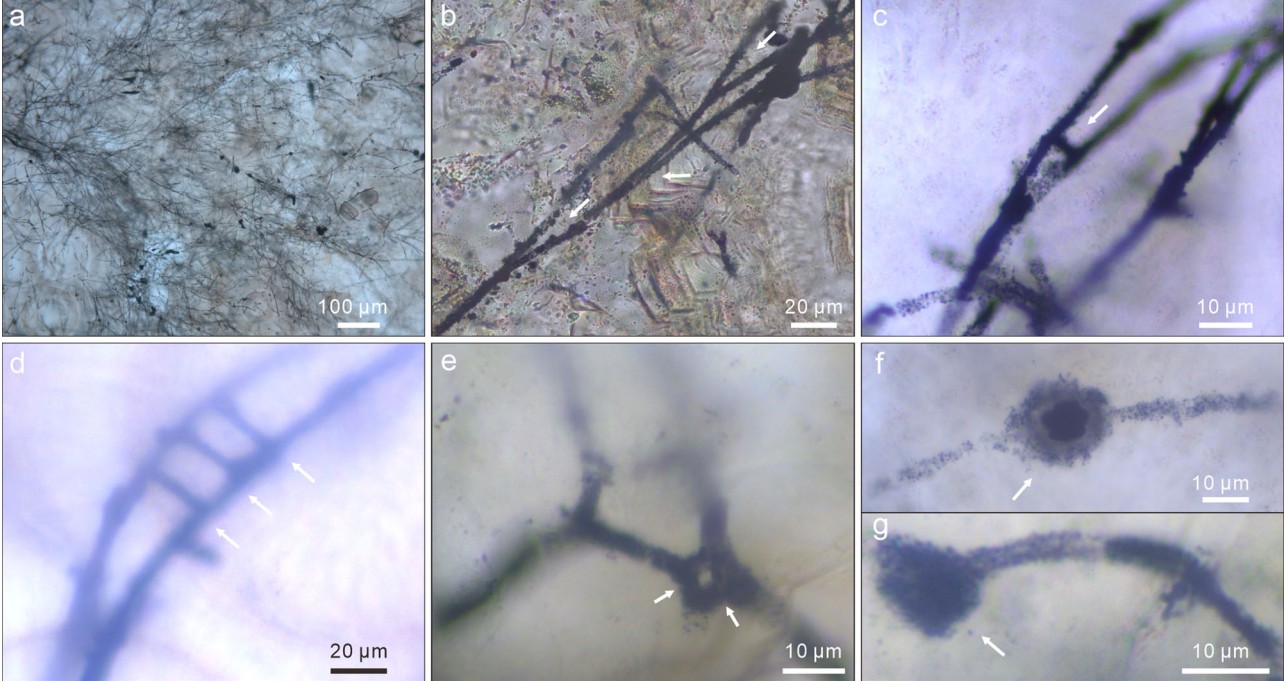

**Fig. 3 TLM photomicrographs of Type B filaments and associated small spheres. a** Aggregate of Type B filaments. **b** Multiple orders of branches (arrows). **c** A-like branching system (arrow). **d, e** Ladder-like or H-like branching systems (arrows). **f** Small intercalary sphere (arrow) coaxially aligned with a filament. **g** Small terminal sphere (arrow) attached to a filament.

product of pyrite. They are similar to the Doushantuo filaments in diameter, preservation, and formation of mycelium-like aggregates, although Zhou et al.[30] stated there is no evidence for branching and anastomosis.

Stigonematalean cyanobacteria (for example, *Fischerella*[31] and *Mastigocladopsis*[32]) can develop true branching filaments with tubular sheaths and differentiated cells such as akinetes and heterocysts[33,34]. However, no stigonematalean cyanobacteria are known to develop aseptate trichome[33]. The extracellular sheaths of filamentous cyanobacteria are more resistant to degradation than cellular trichomes and thus have greater preservation potential[35]. When only the sheaths but not the cellular trichomes are preserved, stigonemataleans can conceivably look like the aseptate Doushantuo filaments. However, neither the sheaths nor the trichomes of stigonematalean filaments can fuse to form A-, H-, or ladder-like branches or networks.

Many eukaryotic algae are also characterized by a true branching organization. However, only a handful of algal groups are characterized by siphonous or siphonocladous filaments that superficially resemble aseptate filaments[36–38]. For example, extant siphonocladaleans (e.g., *Cladophoropsis*[39] and *Rhizoclonium*[40]), vaucheriaceans (e.g., *Vaucheria*[41]), and rhodophyceans (e.g., *Griffithsia*[42]) develop siphonous or siphonocladous thalli[36,37], but again their filaments do not fuse. Some multicellular zygnemataceans (a group of freshwater green algae, for example, *Spirogyra*[43]) can develop H- or ladder-like structures during conjugation (a sexual reproduction process)[36], but their filaments are septate and do not form mycelium-like networks.

Actinobacteria and fungi, both of which can form mycelial networks of branching filaments, are better extant analogs for the Doushantuo filaments than those mentioned above. Actinobacteria can have repeatedly branching filaments that form radial mycelia[44,45], resembling the Doushantuo filaments. Some actinobacteria can have aseptate filaments, and others can produce spores[44,45] that are morphologically comparable to small spheres described here. However, the diameters of these actinobacterial

filaments and spores (~0.15–1.5 μm and ~1 μm in diameter, respectively) are usually smaller than those of the Doushantuo microfossils[9,45]. Importantly, unlike the Doushantuo filaments, actinobacteria characteristically do not form filamentous anastomosis of network[46]. There were rare reports of anastomoses in several strains of the actinobacterial genus *Streptomyces* in the 1950s and 1960s[47,48], but these anastomosal structures remain unconfirmed or they could be derived features within the actinobacteria given their restricted occurrence in the genus *Streptomyces*[47,48].

A better interpretive analog for the Doushantuo filaments is modern fungi, particularly non-Dikarya fungi such as zygomycetes (a paraphyletic group including the Mucoromycota and Zoopagomycota)[49,50]. Unlike the Dikarya (a monophyletic group consisting of Basidiomycota and Ascomycota) that only produces septate filamentous hyphae, zygomycete hyphae are mostly aseptate and can branch monopodially and dichotomously, similar to the Doushantuo filaments (Figs. 2a–h; 3a–e; 4b, g–i, k; Supplementary Fig. 3a–f, j). Cell fusion is a common feature among modern fungi, where filamentous hyphae can fuse to form A-, H-, and ladder-like branches (e.g., in *Neurospora*[51] of the Ascomycota) or interconnected mycelial network (e.g., in nematode-trapping fungi in the Ascomycota such as *Arthrobotrys* and *Dactylella*[12]). Cell fusion, anastomosing hyphae, and mycelial networks also occur in many fungi of the Mucoromycota (e.g., Mortierellomycotina[50]). In addition, sexual reproduction in most zygomycetes also involves cell fusion, the formation of an H-configuration, and the eventual production of zygospores[52]. Thus, when all evidence is considered, the A-, H-, and ladder-like branching systems, as well as the filamentous networks, of the Doushantuo microfossils are best compared with fungal analogs.

In light of a possible fungal affinity of the Doushantuo filaments, the smaller spheres associated with the filamentous hyphae could be interpreted as fungal spores. Similar to the Doushantuo microfossils, modern zygomycetes and many other fungi produce both intercalary and terminal (sometimes chained)

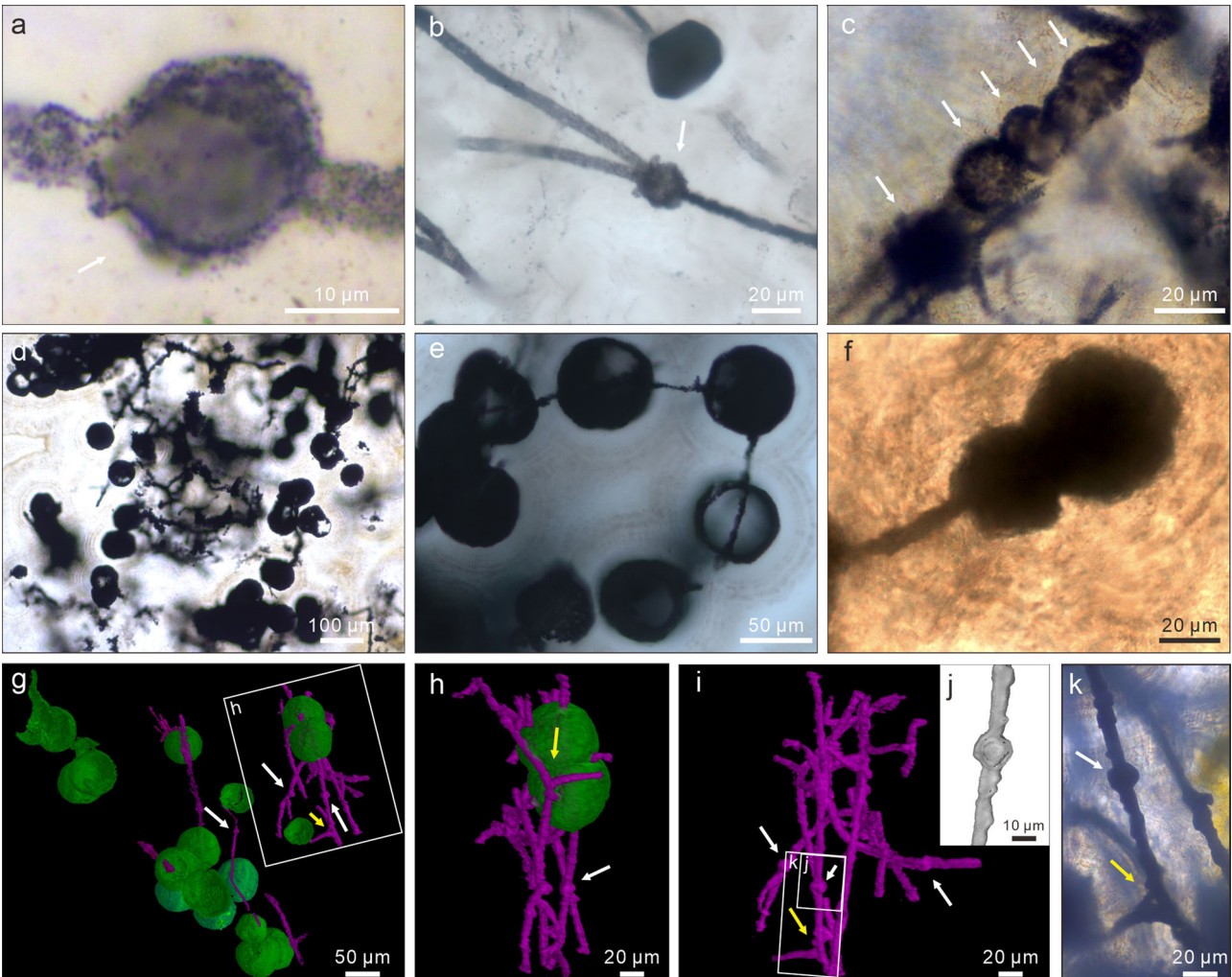

**Fig. 4 TLM photomicrographs, SRXTM surface renderings, and SRXTM cut-away view of spheres associated with Type A filaments. a** Small intercalary sphere (arrow) coaxially aligned with a filament. **b** Small intercalary sphere (arrow) at bifurcation of a branching filament. **c** Small concatenated spheres (arrows) coaxially aligned with a filament. **d** Large spheres. **e** Large spheres coaxially aligned with and penetrated by filaments. **f** Concatenated spheres coaxially aligned with a filament. Terminal sphere is larger. **g–i** SRXTM surface renderings (see Supplementary Movie 2), showing large spheres (green), small spheres (white arrows), filaments (purple), and branching filaments (yellow arrows). **h** corresponds to labeled box in **g**. **j** SRXTM cut-away view of labeled rectangle in **i**, showing hollow nature of small sphere. **k** TLM photomicrograph of labeled rectangle in **i**, showing small sphere (white arrow) and branching filaments (yellow arrow).

chlamydospores that are coaxially aligned with and attached to filamentous hyphae[52]. We note that the large spheres are in tangential contact with and sometimes penetrated by Doushantuo fungus-like hyphae; it is possible that these were symbiotic organisms living together with filamentous fungi, analogous to modern ecto- and endomycorrhizal fungi. It is unclear whether the axial strand in some of the translucent Type A filaments is a bona fide biotic structure, although similar structures have been reported from various fungus-like microfossils[53–55]. Considering that an axial strand is absent in most Doushantuo filaments or in unambiguous fungal fossils[7,10,14], it more likely represents a degradational structure related to the shrinkage of the cellular cytoplasm[55].

Petrographic observations and cross-cutting relationships establish that the Doushantuo microfossils postdate the deposition of the cap dolostone, but predate the chalcedony cement in the sheet-cavities. The microfossils were embedded in the chalcedony, surrounded and truncated by botryoids, and cut by late diagenetic veins. These observations indicate that the micro-organisms were in place before the precipitation of the micro-sparic calcite cement and its subsequent replacement by hydro-thermal chalcedony. A possible sequence of events is as follow (Supplementary Fig. 2). After the cap dolostone was deposited at ~635 Ma, post-glacial isostatic rebound resulted in subaerial exposure, paleo-karstic dissolution, leading to the formation of cryptic sheet-cavities[21]. Carbonate cement, including various speleothems (stalagmites, stalactites, and botryoidal coatings), began to fill the sheet-cavities shortly after their formation[20]. As the sheet-cavities were being filled, fungus-like micro-organisms, probably along with other micro-organisms (considering that the large spheres may represent symbiotic organisms living together with the fungus-like micro-organisms), colonized the cryptic sheet-cavities that were physically connected with a karstic surface atop the ca. 1–4 m-thick cap dolostone (Supplementary Fig. 1e). These micro-organisms were entombed in growing botryoidal cements in the sheet-cavities (Supplementary Fig. 4), analogous to fungi (including zygomycetes) and other micro-organisms found in modern speleothems and ancient karstic

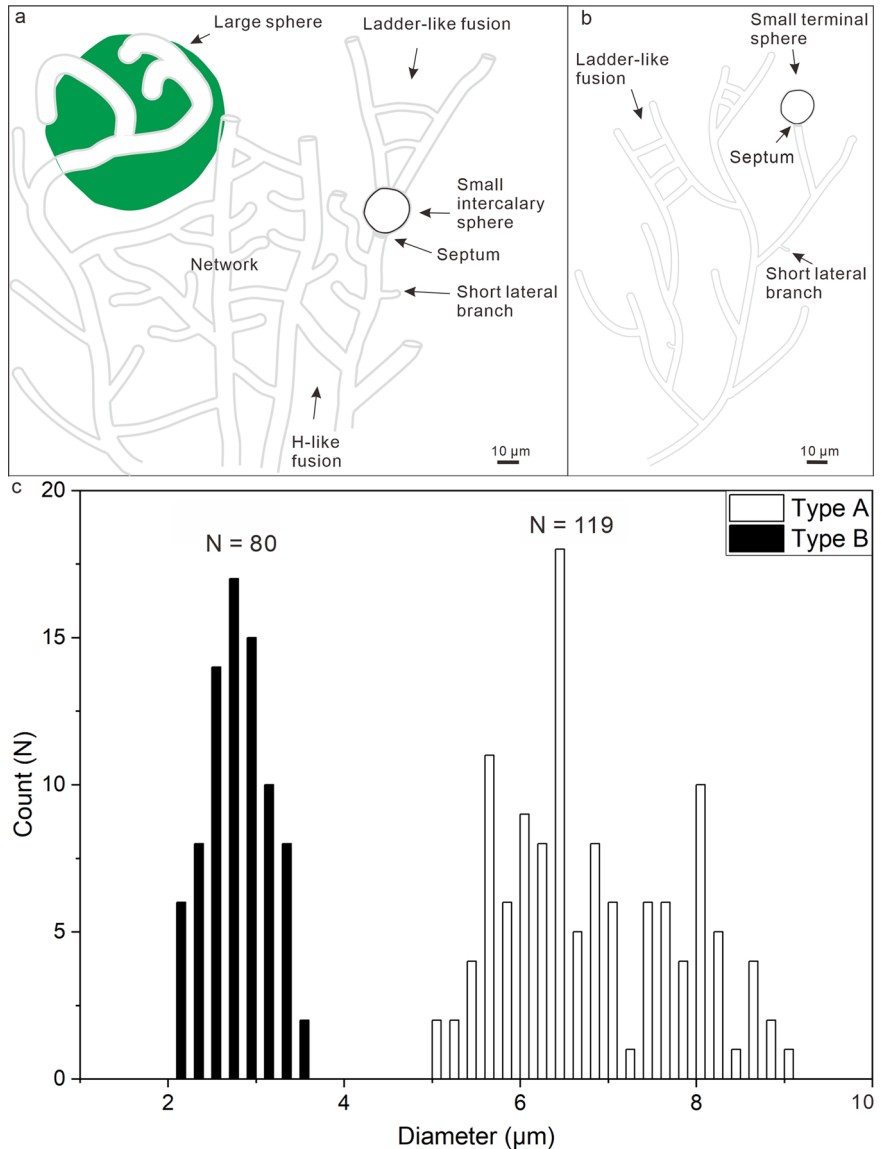

**Fig. 5 Morphological reconstruction and frequency distribution of filament diameter. a, b** Sketches of Type A (**a**) and Type B (**b**) filaments and associated spheres. **c** Frequency distribution of filament diameter of microfossils from Datang. $N = 119$ filaments for Type A filament and $N = 80$ filaments for Type B filament.

caves[56–61] (Supplementary Note 4). Because the sheet-cavities and the cements within occur only in the cap dolostone and do not extend to the overlying strata dated at ~632 Ma[21], it is inferred that the cements formed between 635 Ma and 632 Ma. This inference is consistent with a laser-ablation inductively coupled plasma mass spectrometry (LA-ICP-MS) U-Pb age of $632 \pm 17$ Ma on the isopachous dolomite cement in the sheet-cavities (Supplementary Fig. 8; Supplementary Table 2). Replacement by chalcedony occurred after the precipitation of paleo-karstic speleothems and was probably influenced by hydrothermal activities[22]. Such replacement did not erase the microfossils, which probably had already been pyritized, and enhanced their long-term preservation because silica is more resistant to weathering than calcite and is well known to preserve excellent Precambrian microfossils[62].

In conclusion, our analysis indicates that the Doushantuo filaments likely represent fungal micro-organisms that colonized cryptic karstic environments sometime between 635 Ma and 632 Ma. Indeed, late Ediacaran filamentous fossils reported by Zhou et al.[30] are associated with karstic dolostone and also appear to have some fungal features such as mycelium-like networks and apparently branching filaments (e.g., figure 2a–c, i in ref. [30]); these fossils warrant further investigation to test whether they represent additional Ediacaran examples of fungus-like micro-organisms in cryptic karstic environments. Regardless, the Doushantuo microfossils reported here extend the fossil record of putative terrestrial fungi[15] by 220 Myr and predate the earliest embryophytes by >100 Myr[63]. Together with other terrestrial microbes that likely included cyanobacteria and green algae[64–66], these fungus-like micro-organisms fostered a relatively simple terrestrial ecosystem in the aftermath of the terminal Cryogenian snowball Earth glaciation. If proven to be ecologically widespread, these terrestrial microbes could accelerate chemical weathering and the delivery of phosphorus into the ocean[67], thus stimulating marine bioproductivity. They could also facilitate the production of detrital clay minerals[2], which play a key role in organic carbon sequestration. Together, elevated marine bioproductivity coupled with greater efficiency of organic carbon sequestration means

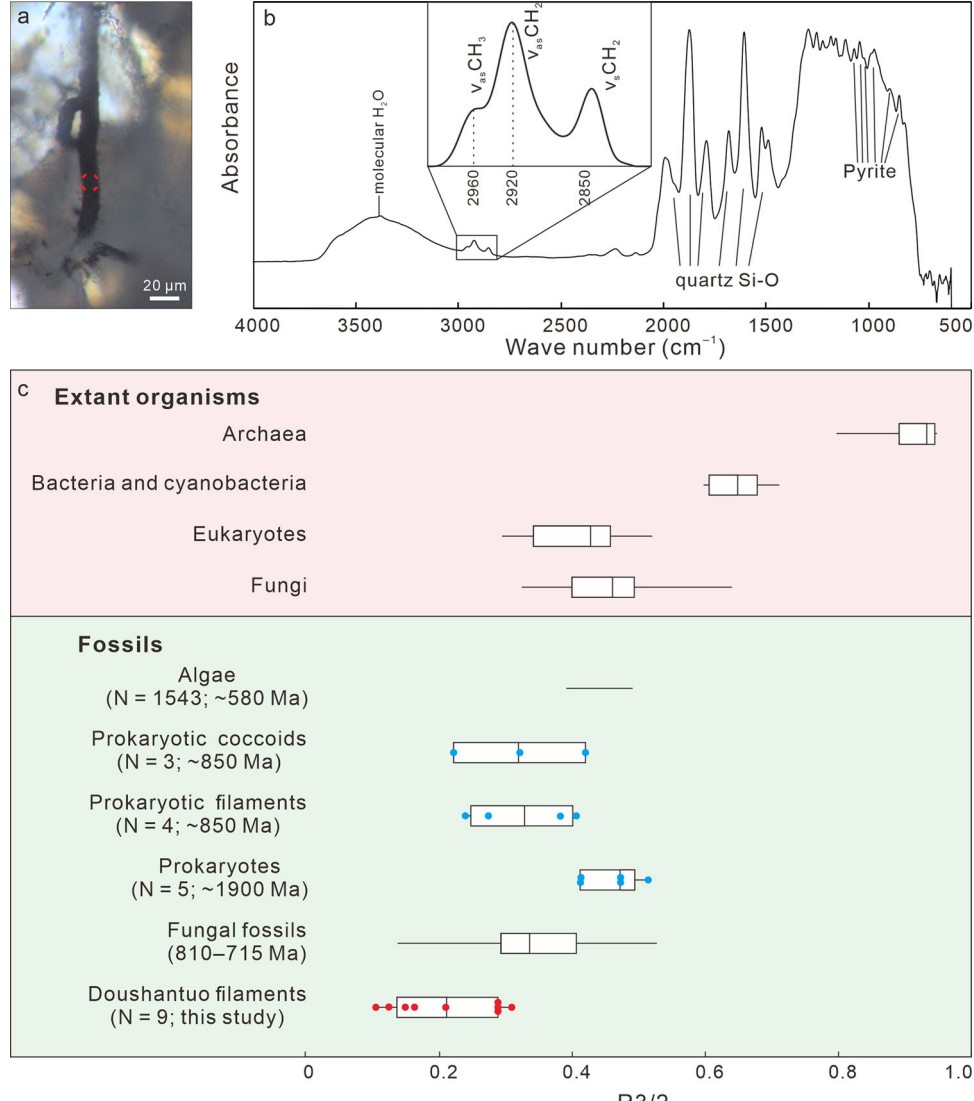

**Fig. 6 FTIR of Type A filament from Beidoushan. a** TLM photomicrograph of Type A filament, with red circle denoting spot of FTIR analysis shown in **b**. **b** FTIR spectrum, with enlargement of 2800–3000 cm$^{-1}$ spectral region (inset). Bands at 2960 cm$^{-1}$, 2920 cm$^{-1}$, and 2850 cm$^{-1}$ are due to asymmetric aliphatic CH$_3$ (end-methyl), asymmetric, and symmetric aliphatic CH$_2$ (methylene-chain), respectively[10,11]. Relative intensities of 2960 cm$^{-1}$ and 2920 cm$^{-1}$ bands were used to calculate R3/2 ratios that reflect ratios of -CH$_3$ to -CH$_2$ groups[11]. Band at ~3400 cm$^{-1}$ is due to molecular water[10]; bands at 1992, 1875, 1792, 1683, 1608, 1522, and 1490 cm$^{-1}$ are due to Si–O bonds of quartz[81]; bands at 1071, 1045, 1024, 974, 852, and 826 cm$^{-1}$ are probably due to pyrite[82]. **c**, Box-and-whisker plot of R3/2 ratios of representative modern organisms and fossils. Plots for extant archaea, prokaryotes, eukaryotes, and fungi, as well as 810–715 Ma fungal fossils are from ref. [11] and references therein. Data for fossil algae (N = 1543 spot analyses in three specimens) are from ref. [83]. Data for fossil prokaryotic coccoids (blue dots; N = 3 specimens), prokaryotic filaments (blue dots; N = 4 specimens), and prokaryotes (blue dots; N = 5 specimens) are measured from fig. 5a of ref. [81] using ImageJ 1.47 v. Data for Doushantuo filaments (red dots; N = 9 spot analyses in two filaments) are based on analyses of sample 18BD-25 from Beidoushan. Box-and-whisker plots show the median (central line), the minimum and maximum (whiskers), and the 25th–75th percentile (bounds of the box), whereas horizontal line for fossil algae show the minimum and maximum. Source data are provided as a Source Data file.

enhanced organic carbon burial and resultant atmospheric-oceanic oxygenation at ~635–630 Ma[68]. Thus, the Doushantuo fungus-like micro-organisms, as cryptic as they were, may have played a role in catalyzing atmospheric oxygenation and biospheric evolution in the aftermath of the terminal Cryogenian global glaciation.

## Methods
**Studied material**. Nine sheet-cavity samples (16DT-2, 16DT-2b, 16DTC-2A, 17DT-A-1, 17DT-A-2, 17DT-A-4, 17DT-A-6, 17DT-A-9, and 17DT-2A) were collected from the cap dolostone of the Doushantuo Formation at the Datang section (27°01′54.5″N, 107°24′08.4″E) and eleven sheet-cavity samples (18BD-6, 18BD-8, 18BD-9-1, 18BD-12, 18BD-20b, 18BD-25, 18BD-33, 18BD-32, 18BD-34,

18BD-35, and 18BD-36) were collected from the cap dolostone of the Doushantuo Formation at the Beidoushan section (27°01′40.4″N, 107°23′22.0″E), Weng'an, Guizhou Province, South China (Fig. 1; Supplementary Fig. 1). Petrographic thin sections of various thicknesses (30 μm, 100 μm, 200 μm, and 500 μm) and polished slabs of the sheet-cavity samples were prepared with controlled stratigraphic orientations (perpendicular to or parallel with bedding plane).

**Light microscopy**. Thin sections were examined and photographed on a Leica DM4500P microscope coupled with a Nikon D750 digital camera at the Institute of Geochemistry, Chinese Academy of Sciences, a Zeiss Axioscope A1 microscope with a Axiocam 506 digital camera at the Nanjing Institute of Geology and Palaeontology, Chinese Academy of Sciences, a Zeiss Axioscope A1 microscope with a Axiocam 512 digital camera at Virginia Tech, an Olympus SZ16 microscope with a DP27 digital camera at Virginia Tech, and an Olympus BX-60 upright

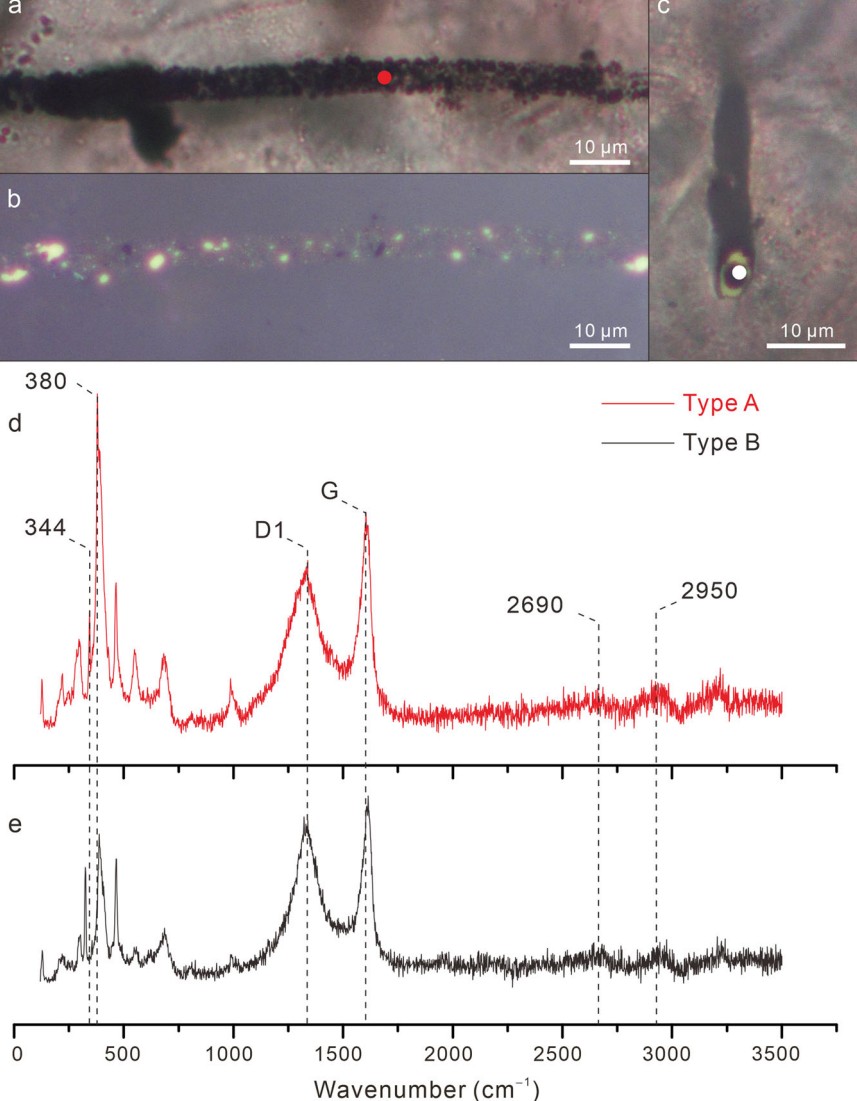

**Fig. 7 Light microscopic images and Raman spectra of Type A and Type B filaments. a, b** TLM (**a**) and reflected light microscopy (RLM) (**b**) photomicrographs of the same Type A filament. **c** TLM photomicrograph of Type B filament. **d, e** Raman spectra of Type A and Type B filaments acquired on locations marked by circular dots in **a** and **c**, respectively, showing characteristic bands for low-grade carbonaceous material, including D1-band at ~1350 cm$^{-1}$ and G-band at ~1580 cm$^{-1}$ in the first order region (1100–1800 cm$^{-1}$), and bands at ~2700 and ~2900 cm$^{-1}$ in the second-order region[84]. Raman bands at ~340 and ~380 cm$^{-1}$ are characteristic of pyrite (FeS$_2$)[85]. Both analyzed filaments are from Datang (sample 17DT-A-9).

microscope with an Olympus SC30 digital camera at the University of Cincinnati. Digital images were not processed other than brightness and contrast optimization and image stacking. Frequency distribution of filament diameter was prepared using the software Microsoft Excel 2013.

**SRXTM**. SRXTM analysis was performed using the 13 W beam line at the SSRF by propagation phase-contrast synchrotron X-ray microtomography (PPC-SR X-ray μCT) at the Shanghai Synchrotron Radiation Facility. The detector was Binson Flash 4.0 with a pixel size of 6.5 μm and 2048 × 2048 pixels. The distance between the sample and the detector was 10 cm, and the projection was 1600 on 180°. The synchrotron radiation beam was monochromatized at 15 KeV by a two-crystal monochromator. The PITRE V3.1 software was used for phase retrieval and slice reconstruction of the data, and ImageJ 1.47 v was used for rotation correction. Three-dimensional reconstruction was performed using VG Studio 2.0 (Volume Graphics, Heidelberg, Germany) software.

**CLSM**. CLSM imaging was performed using an Olympus Fluoview 1200 CLSM with the software package FV10-ASW (v.3.01) at the University of Cincinnati, following established procedures[23,69,70]. Three-dimensional images were acquired by use of 488 nm laser excitation at 100% transmission (~300 μW at the sample), a ×60 oil-immersion objective (numerical aperture = 1.42) with fluorescence-free

microscopy immersion oil (Olympus Type-F), and a 505–605 nm bandpass filter to exclude the incident laser wavelength.

**FTIR spectroscopy**. FTIR spectra were acquired using a Fourier transform vacuum infrared micro-spectrometer (Vertex-70V, Hyperion-1000 infrared microscope) at the Institute of Geochemistry, Chinese Academy of Sciences. Each IR (infrared spectroscopy) absorption measurement was conducted with unpolarized radiation from a mid-IR light source, a mercury-cadmium-telluride detector with a 100 × 100 μm aperture, and a CaF$_2$ beam splitter. Background was collected on a ZnSe plate free of sample. Each IR spectrum in the wavenumber range of 4000–500 cm$^{-1}$ was collected with 100 accumulated scans. Baseline correction and peak height measurements were carried out using the OMNIC 8.0 software.

**Raman spectroscopy**. Selected microfossils were analyzed using a laser Raman micro-spectrometer (Renishaw inVia Reflex, UK) equipped with a charge-coupled device multi-channel detector and a Leica microscope at the Institute of Geochemistry, Chinese Academy of Sciences. The source was a Spectra-Physics argon ion laser (λ = 514.5 nm; 20 mW in maximum power) with a 1800 grooves/mm grating. The laser was focused to a spot ≤1 μm in diameter with a ×100 objective lens to collect backscattered radiation. The measured Raman shift was between 120 and 3500 cm$^{-1}$. Raman mapping of pyritized microfossils was acquired at the University of Cincinnati using a Horiba T64000 Raman system (Horiba, Inc.,

Edison, NJ) with an Olympus BX41 microscope equipped with an X-Y-Z motorized stage capable of 0.1 μm steps in all three dimensions. A ×50 objective lens (numerical aperture = 0.50) was used to focus the 457.9 nm excitation from a Coherent FreD 90 C Ar$^+$ laser to a spot size of ~1–2 μm. Data were collected and processed using the software LabSpec (v.5; Horiba, Inc., Edison, NJ). Raman spectra for the map were collected in a 40 × 23 μm X-Y grid with 1 μm spacing between points. Each spectrum was acquired for a total of 1 s (two 0.5 s acquisitions per spectrum to allow the software to remove cosmic ray spikes). Laser power at the sample was ~8 mW. The map was produced by integrating the intensity of the pyrite band between 355 and 405 cm$^{-1}$.

**Electron microscopy.** FIB-SEM analysis was performed at the Institute of Geochemistry, Chinese Academy of Sciences, using a FEI Scios Dual beam that combines a traditional Field Emission electron column with a FIB column equipped with EDS, ETD (secondary electron, SE), and T1 (backscattered electron, BSE) detectors. In situ ion-milled lift-out procedures were performed by FIB-SEM to prepare ultrathin foils (<100 nm thick) for scanning transmission electron microscopy observation[71]. Ultrathin foils of the filamentous microfossils were then observed under a FEI G2 F20 S-TWIN transmission electron microscope (at 200 keV) at the State Key Laboratory of Mineral Deposits Research, Nanjing University. Elemental compositions of the microfossils and matrix were detected on an OXFORD EDS X-Max 80 T instrument. Scanning electron microscopy (SEM) imaging was performed using a Phenom ProX equipped with BSE and EDS detectors.

**Nanoscale SIMS.** In situ nanoscale SIMS (Nano-SIMS) analysis was performed on a Cameca Nano-SIMS 50 L instrument at the Institute of Geology and Geophysics, Chinese Academy of Sciences. Pyrite sulfur isotope data were acquired by rastering areas ranging from 20 × 20 μm$^2$ to 40 × 40 μm$^2$ using a Cs$^+$ beam of 7–10 pA with a diameter of ~250 nm. The $^{34}$S/$^{32}$S ratios ($^{32}$S was counted with a Faraday cup and $^{34}$S was counted with an electron multiplier) were measured in a spot analysis mode, by rastering the same current over an area of 2 × 2 μm$^2$. The $^{34}$S/$^{32}$S ratios were calibrated for matrix effects with the Sonora pyrite standard (for more detailed analytical method and calibration standard, see refs. [72,73]). δ$^{34}$S$_{pyrite}$ data are reported as ‰ deviation from V-CDT and presented in Supplementary Table 1.

**LA-ICP-MS.** U-Pb radiometric dating of carbonate using LA-ICP-MS, including the standards NIST 614 and WC-1 for normalization of $^{207}$Pb/$^{206}$Pb and $^{238}$U /$^{206}$Pb ratios, was described previously[74,75]. U-Pb isotopes were measured in situ on isopachous dolomite in sheet-cavities from the Daping section (Hunan Province of South China) on an Agilent 7900 quadruple mass spectrometer equipped with an inductively coupled plasma and a GeoLasPro 193 nm ArF excimer laser (LA-ICP-MS) at the State Key Laboratory of Ore Deposit Geochemistry, Institute of Geochemistry, Chinese Academy of Sciences. Samples were ablated by a laser of 120 μm spot size, 10 Hz repetition, and 8 J cm$^{-2}$ energy density. The aerosols were carried by pure Helium (450 ml/min) and then mixed with argon via a T-connector and finally injected to the ICP torch. Pure nitrogen (~3.0 ml/min) was added to the helium carrier gas to increase the sensitivity via a Y-junction before entering ICP[76]. Samples were pre-ablated for 3–5 pulses before analysis to remove surface contaminations. Each analysis included 20 s for background acquisition, 30 s for data acquisition, and 40 s for elimination of memory effects. The approach of standard-sample bracketing was used to measure the isotopes of 202, 204, 206, 207, 208, 232, and 238. Data reduction was undertaken off-line using ICPMSDataCal (v.11)[77] and then regressed to determine a discordia line on Tera-Wasserburg model using Isoplot 3.0. U-Pb isotopic data are presented in Supplementary Figure 8 and Supplementary Table 2.

**Reporting summary.** Further information on research design is available in the Nature Research Reporting Summary linked to this article.

## Data availability
The data that support the findings of this study are available in the paper and its supplementary information files, or from the corresponding authors upon reasonable request. All specimens illustrated in this paper are reposited in Nanjing Institute of Geology and Palaeontology (NIGPAS, Nanjing, China), with NIGPAS museum catalog numbers (prefix PB-) given for illustrated specimens (see Supplementary Table 3 for repository information). Source data are provided with this paper.

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

## Acknowledgements

This research was supported by the National Natural Science Foundation of China (41530210, 41873058, and 41921002), Natural Science Fund of Guizhou Province, China [JZ(2015)2009], Chinese Academy of Sciences (XDB 26000000, QYZDJ-SSW-DQC009, and XDB18000000), and National Key Research and Development Program of China (2017YFC0603100). T.G. acknowledges financial support from China Scholarship Council. S.X. acknowledges support from the US National Science Foundation (EAR-2021207). K.P. acknowledges support from the Youth Innovation Promotion Association of the Chinese Academy of Sciences. We thank Xunlai Yuan for discussion, Xingzhong Liu, and Xiaoyong Liu for the information of extant fungi, Jordan Metzgar at Virginia Tech Massey Herbarium for providing extant fungal specimens for comparison, and Yue Zhao, Michelle Stocker, and Chris Griffin for SXRTM data processing.

## Author contributions

T.L., K.P., and T.G. designed the research. T.L., G.Z., and T.G. collected samples. T.G., T.L., K.P., A.D.C., and S.X. conducted the light microscopic analysis. A.D.C. conducted CLSM observation. T.G. conducted other experiments with help from G.L. and Q.Y. in in-line phase-contrast Synchrotron Radiation X-ray microtomography and 3D modeling. S.X., T.G., K.P., and T.L. developed the interpretation. K.P., T.G., S.X., and T.L. took the lead in writing the manuscript with contributions from C.Z. and B.W.

## Competing interests

The authors declare no competing interests.

**Additional information**

