## [Peer Review File · Nature Communications]

Reviewers' Comments:

Reviewer #1:

Remarks to the Author:

Fossils are important in part because they show character combinations that cannot be inferred or predicted from phylogenetic analysis of modern organisms. Gan et al. report on 635 Ma filaments that they interpret as probable fungi. There are few places in the world where terrestrial fossils of this great age are preserved, and so finding the filaments adds significantly to the overall picture of life in deep time.

Based on analysis of modern organisms, the fossils reported in this manuscript, like the filaments that Ivarsson et al. have described from marine basalt, are deeply puzzling. Excellent mysteries are worth publishing. However, from my perspective, admittedly rooted in analysis of extant fungi, the fossils more likely represent sheathed prokaryotes or other organisms rather than fungi.

Why interpretation as fungi seems unlikely

1. The fossils were in habitats without light, without photosynthetic organisms presumably. Genomic comparisons suggest that the ancestral fungal secretome was directed at extracting nutrition from plant cell walls. It is particularly unlikely that early zygomycetes would be found in this habitat, since among all fungi, modern zygomycete phyla are most limited in their digestive repertoire. They are adapted to readily accessible, nutrient rich environments.

2. The references to modern speleothems added in revision tend to raise questions. Is there evidence that fungi in caves could grow and reproduce using carbon from prokaryotes or other carbon sources available 635 Ma in speleothems?

Line 197: "These micro-organisms were embedded in growing botryoidal cements in the sheet-cracks (Extended Data Fig. 5), analogous to fungal and other micro-organisms found in modern speleothems³⁵⁻³⁷."

Content of the cited references does not support idea that filaments in 635 Ma caves were likely to have been fungi.

Ref 35, Vaughn et al 2015. Reports that fungal diversity detected as metagenomic DNA might be related to propagules that enter via a cave opening (not necessarily organisms that were growing in the cave.) The proportion of zygomycetes was very low away from the cave entrance (suggesting that propagules of zygomycetes came in but were not reproducing in the cave. This is consistent with zygomycete requirements for abundant, easily accessible nutrients.) Some of the more common species of fungi eg *Cryptococcus* are related to cave animals that were not present 635 Ma.

Ref. 37. This reference was an analysis of cave bacteria, using prokaryote specific primers. There is a statement 'Rare fungal hyphae were observed, suggesting that fungi do not comprise a significant portion of the moonmilk microbial communities.'

Ref 77 Mendoza et al. 2016. The fungi detected as DNAs include obligate plant pathogens, that could not be living or growing in caves. The fact that many of them are plant pathogens supports the suggestion of DNA leaching in the speleothem. Other fungi detected could also have been from leachate.

Why bacteria are difficult to rule out.

3. Actinomycetes, at least some species, do anastomose and can make ladder-like patterns of anastomosis (Gregory, 1956, HYPHAL ANASTOMOSIS AND CYTOLOGICAL ASPECTS OF STREPTOMYCES-SCABIES). The report is old but Gregory's micrographs and the evidence from staining look convincing and are supported by a couple other papers from the same era. Detailed morphological analysis of prokaryotes is a thing of the past and I found no updates. Even if Gregory was wrong, his images show structures that would be difficult to distinguish from anastomosis.

4. Most actinomycetes species are less than 2 μm wide at their widest (ie, Bergey's manual, descriptions of species). But sheaths may result in filaments that are wider and it remains possible that the Doushantuo fossils are mostly sheaths. Would wide bacteria be more probable than early fungi adapted to non-algal carbon sources?

5. Lines 147-149. Citation 7 (Bengtson 2017) is inappropriate here. It's necessary to look for original observations.

6. Bacteria also have n-acetylglucosamine in their sheaths or peptidoglycan walls, which can explain the detection of chitin-like biochemistry with lectins and maybe also, after modification through time, with μFTIR .

The central core needs to be discussed.

7. The description of the central core in the filaments seems to be missing from the revised text (was line 142-183 in the previous version). The central core is clearly visible in images 1 c, d, e. The explanation in the rebuttal raises more questions. If the thick outer layer is debris, then the actual diameter of the living cell is the diameter of the central mold. The central mold is thin or very thin, not the diameter of filaments of type 'A'.

8. The thin central strands in images 1 c, d, e, and also in fossils from Ivarsson, are not found in bona fide fossils of fungi.

9. Could the filaments be interpreted as sheaths? What is known about deposition of silica or carbonates in sheaths or around filaments?

10. If only the central strands in Fig. 1c etc are from the organism, what's the evidence that single, outer layer is not also debris, if it is similar in diameter and morphology to 'debris' shown in Fig. 1c?

11. Is it possible to assess cytoplasmic anastomosis if only outer debris can be observed?

Other minor points

12. Line 39. The other refs are fine, but 'the Middle Ordovician Guttenberg Formation in Wisconsin17' probably represents contamination (see book, Fossil Fungi, 2015 by Taylor et al). Evidence of synchronicity is not presented and the cracked dolomite outcrop likely allowed penetration of more recent roots.

13. Line 100. 1a nor 3c curve but they are not 'superficially resembling clamp connections of the Basidiomycota' unless any curving biological side branch superficially resembles a clamp. In 1a, the curved branch is growing past the maternal filament, unlike a clamp tip that would fuse with the maternal filament. In 3c, the filament is growing away from the maternal hypha before hooking. The lack of septa rules out clamps. Consider deleting, or add an explanation about why, as the authors correctly recognized, the hook can't be considered part of a clamp.

14. lines 120-121. Clarify what it is about the EDS elemental maps that indicate organic matter, not just pyrite. FTIR and Raman are, however, convincing.

15. Line 160. Wording/cladistics. No extant organisms, zygomycetes included, are 'lower' in a phylogenetic sense. It's necessary to consider, not the characters of modern zygomycetes, but rather traits in the common ancestor of zygomycetes and their sister clades.

16. lines 178-180. Almost all fungi, and possibly also some bacteria produce intercalary 'chlamydospores.' The significance of these structures isn't easy to interpret. Although the name suggests that they are spores, whether they have spore-like function is unclear.

17. Line 180. No basis for 'symbiosis'. For 'symbiosis', it would, at the least, be necessary to demonstrate that both organisms had been alive at the same time. The filaments could be saprotrophic.

18. Line 184-209. Clarify how microbes could have entered the sheet cracks. Supplementary Fig. 1 suggests that the cracks were deep below ground. The added text explains more about possible paths to fossilization, but does not discuss how microbes found their way into caves.

19. Extended Fig. 6. Should present more available data from fossils, including cyanobacteria and other prokaryotes. The Doushantuo filament ratios are significantly different from all extant organisms and this suggests that R3/R2 changes with diagenesis.

20. Extended Fig. 8. Add more explanation to the legend to aid in interpretation. What does it show about the nature of the remains of the fossil? Does it just show that the filaments are pyritized? Specify the location of the fossil, relative to the matrix in images e through p?

21. Extended Fig. 9. Again, add interpretation to legend.

In summary, the fossils are interesting. The authors have presented good arguments for excluding modern algal groups but the edits to strengthen the justification for a fungal affinity are not convincing. The problems with a fungal interpretation should be confronted.

Reviewer #2:

Remarks to the Author:

The manuscript by Pang and colleagues reports fungal microfossils preserved in sheet-cracks from the Ediacaran Doushantuo Formation in China. This is a work of importance for the understanding of late Precambrian biosphere as well as for the processes that might have led to terrestrialisation of eukaryotes. I recommend this work for publication. During my previous review of the manuscript I raised some concerns with the comparisons for biological affinities, the taxonomical nomenclature and the origin of these fungal organisms. The authors have certainly addressed all of these concerns in the present submission.

Reviewer #3:

Remarks to the Author:

The authors have in a satisfactory way responded to the suggestions from the reviewers. In my opinion, the article is now almost fit for publication. The only thing I'm missing is information about the composition of the large spheres, given that their relationship to the filaments and the small spheres is not wholly clear. A complement to the EDS maps of a small sphere (Extended Data Figure 9) to included corresponding data for large spheres would be informative.

One minor point: In the abstract (line 26) the interpretation of the fossils as "eukaryotes and probable fungi" is preceded (line 23) by a more definitive "fungal microfossils". If line 23 is revised to "fungal-like microfossils" the text will be more consistent.

REVIEWER COMMENTS

Reviewer #1 (Remarks to the Author):

Fossils are important in part because they show character combinations that cannot be inferred or predicted from phylogenetic analysis of modern organisms. Gan et al. report on 635 Ma filaments that they interpret as probable fungi. There are few places in the world where terrestrial fossils of this great age are preserved, and so finding the filaments adds significantly to the overall picture of life in deep time.

Based on analysis of modern organisms, the fossils reported in this manuscript, like the filaments that Ivarsson et al. have described from marine basalt, are deeply puzzling. Excellent mysteries are worth publishing. However, from my perspective, admittedly rooted in analysis of extant fungi, the fossils more likely represent sheathed prokaryotes or other organisms rather than fungi.

Why interpretation as fungi seems unlikely

1. The fossils were in habitats without light, without photosynthetic organisms presumably. Genomic comparisons suggest that the ancestral fungal secretome was directed at extracting nutrition from plant cell walls. It is particularly unlikely that early zygomycetes would be found in this habitat, since among all fungi, modern zygomycete phyla are most limited in their digestive repertoire. They are adapted to readily accessible, nutrient rich environments.

Response: We appreciate the reviewer's concern about nutrient availability. We would like offer the following responses. **First**, considering that the sheet-cracks are only 1–4 meters below the karstic surface atop the Doushantuo cap dolostone, photosynthetic organisms may have lived in these cracks where light was accessible through rock fissures. Behrendt et al. (2020), for example, has shown that red light can reach deep caves through mineral-dependent reflectance. Indeed, cyanobacteria are common in modern caves, sometimes over 100 m deep in limestone caves (Czerwik-Marcinkowska et al., 2015; Popović et al., 2015). So it is not totally inconceivable that cyanobacteria may have colonized the sheet-cracks in the Doushantuo cap dolostone. **Second**, nutrients produced by photoautotrophs on the surface of the landscape can be transported into the sheet-cracks by groundwater. **Third**, although limited light and low nutrient conditions in underground caves

present unique challenges for potential colonizers, modern and fossil ecosystems have been documented in very deep caves (e.g., more than one hundred meters beneath the entrance)(Northup et al., 1994; Nováková, 2009). For example, zygomycetes and other fungal taxa have been reported from the Lechuguilla Cave (Northup et al., 1994) and mucoralean fossils have been reported from in ancient karstic caves (Kretzschmar, 1982). **Finally**, freshwater streptophytes likely diverged before the Ediacaran Period. It has been suggested that the divergence of streptophytes, which contains pectin- and hemicellulose, provides a maximum age constraint on the divergence of osmotrophic fungi (Berbee et al., 2020; Chang et al., 2015). Several molecular clock studies suggest that streptophytes diverged in the Tonian or Cryogenian (Cortona et al., 2019; Morris et al., 2018). In addition, the finding of 1,000-million-year-old crown-group chlorophyte fossils (Tang et al., 2020) also indicates that chlorophytes and streptophytes must have diverged before the Ediacaran Period. Therefore, there are no *a priori* reasons to exclude the possibility of Ediacaran zygomycetes.

2. The references to modern speleothems added in revision tend to raise questions. Is there evidence that fungi in caves could growth and reproduce using carbon from prokaryotes or other carbon sources available 635 Ma in speleothems?

Response: There has been very little research on the nutrient sources for modern cave fungi. In principle, nutrients for Doushantuo fungi in karstic environment may have come from mixed carbon sources, including both prokaryotes and eukaryotes. The large spheroidal microfossils that are preserved in association with the Doushantuo fungi may represent autotrophs and could have been a nutrient source. In addition, as discussed above, cyanobacteria and green plants on the surface of the landscape may have generated organic carbon that was then transported through groundwater to sheet-cracks in the Doushantuo cap dolostone.

Line 197: "These micro-organisms were embedded in growing botryoidal cements in the sheet-cracks (Extended Data Fig. 5), analogous to fungal and other micro-organisms found in modern speleothems 35-37."

Content of the cited references does not support idea that filaments in 635 Ma caves

were likely to have been fungi.

Ref 35, Vaughn et al 2015. Reports that fungal diversity detected as metagenomic DNA might be related to propagules that enter via a cave opening (not necessarily organisms that were growing in the cave.) The proportion of zygomycetes was very low away from the cave entrance (suggesting that propagules of zygomycetes came in but were not reproducing in the cave. This is consistent with zygomycete requirements for abundant, easily accessible nutrients.) Some of the more common species of fungi eg *Cryptococcus* are related to cave animals that were not present 635 Ma.

Ref. 37. This reference was an analysis of cave bacteria, using prokaryote specific primers. There is a statement 'Rare fungal hyphae were observed, suggesting that fungi do not comprise a significant portion of the moonmilk microbial communities.'

Ref 77 Mendoza et al. 2016. The fungi detected as DNAs include obligate plant pathogens, that could not be living or growing in caves. The fact that many of them are plant pathogens supports the suggestion of DNA leaching in the speleothem. Other fungi detected could also have been from leachate.

Response: Refs. 35–37 and 77, which report fungal DNAs from caves, have been replaced by references that report fungi and other micro-organisms from caves based on in situ observations or laboratory cultures (Nieves-Rivera et al., 2009; Northup et al., 1994; Nováková, 2009; Popović et al., 2015) (Line 233–236).

The reviewer points out that terrestrial organic carbon (e.g., algae and plants) can be leached and transported to caves, thus providing a nutrient base to support a cave ecosystem. This partly addresses the reviewer's comment #2 above.

Revised text (Line 233–236):

“These micro-organisms were entombed in growing botryoidal cements in the sheet-cracks (Supplementary Fig. 4), analogous to fungi (including zygomycetes) and other micro-organisms found in modern speleothems and ancient karstic cave⁵³⁻⁵⁸ (Supplementary Information).”

Why bacteria are difficult to rule out.

3. Actinomycetes, at least some species, do anastomose and can make ladder-like patterns of anastomosis (Gregory, 1956, HYPHAL ANASTOMOSIS AND CYTOLOGICAL ASPECTS OF STREPTOMYCES-SCABIES). The report is old but Gregory's micrographs and the evidence from staining look convincing and are supported by a couple other papers from the same era. Detailed morphological analysis of prokaryotes is a thing of the past and I found no updates. Even if Gregory was wrong, his images show structures that would be difficult to distinguish from anastomosis.

Response: There are a few isolated and old reports of anastomoses in a few strains of the actinobacterial genus *Streptomyces* (Gregory, 1956; Higgins and Silvey, 1966). However, similar structures have not been confirmed in subsequent investigation of *Streptomyces*, which is one of the most speciose (> 500 species described) and most studied genera of actinobacteria. It is possible that the stains used in Gregory's study may have stimulated the formation of anastomoses. It is also possible that the "anastomoses" were not permanent or an optical artifact of overlapping filaments (Higgins and Silvey, 1966). These anastomosal structures remains unconfirmed in *Streptomyces*, or they represent a derived character of *Streptomyces* given that most *Streptomyces* strains do not form anastomoses (Goodfellow et al., 2012).

Revised text (Line 183–188):

"Importantly, unlike the Doushantuo filaments, actinobacteria characteristically do not form filamentous anastomosis of network⁴³. There are rare reports of anastomoses in several strains of the actinobacterial genus *Streptomyces* in the 1950s and 1960s^{44,45}, but these anastomosal structures remains unconfirmed or they could be derived features within the actinobacteria given their restricted occurrence in the genus *Streptomyces*^{44,45}."

4. Most actinomycetes species are less than 2 µm wide at their widest (ie, Bergey's manual, descriptions of species). But sheaths may result in filaments that are wider and it remains possible that the Doushantuo fossils are mostly sheaths. Would wide bacteria be more probable than early fungi adapted to non-algal carbon sources?

Response: We feel that the Doushantuo filaments are unlikely actinobacterial sheaths. The main reason is that H- and ladder-like branching systems, as well as mycelium-like networks, of the Doushantuo filaments indicate cell fusion, which is a mechanism of intercellular communication and is less likely, if not impossible, between cells enclosed within a thick sheath that would impede such communications. As discussed above, although there are isolated and old reports of anastomoses in a few strains of *Streptomyces* (Gregory, 1956; Higgins and Silvey, 1966), these reports have not been confirmed in more recent studies. More importantly, *Streptomyces* does not have a sheath and its filaments are less than 2 μm wide (Goodfellow et al., 2012).

See response to comments #1-2 with regard to the possible nutrient sources for the Doushantuo fungus-like micro-organisms.

5. Lines 147-149. Citation 7 (Bengtson 2017) is inappropriate here. It's necessary to look for original observations.

Response: Original references have been added and the sentence has been revised accordingly (Line 181–188).

Revised text (Line 181–188):

“However, the diameters of these actinobacterial filaments and spores ($\sim 0.15\text{--}1.5\ \mu\text{m}$ and $\sim 1\ \mu\text{m}$ in diameter, respectively) are usually smaller than those of the Doushantuo microfossils^{9,42}. Importantly, unlike the Doushantuo filaments, actinobacteria characteristically do not form filamentous anastomosis of network⁴³. There are rare reports of anastomoses in several strains of the actinobacterial genus *Streptomyces* in the 1950s and 1960s^{44,45}, but these anastomosal structures remains unconfirmed or they could be derived features within the actinobacteria given their restricted occurrence in the genus *Streptomyces*^{44,45}.”

6. Bacteria also have n-acetylglucosamine in their sheaths or peptidoglycan walls, which can explain the detection of chitin-like biochemistry with lectins and maybe also, after modification through time, with μFTIR .

Response: Thanks for this information. In this study, the μ FTIR data provide evidence of the preservation of organic matter, but no direct evidence for the preservation of chitin.

The central core needs to be discussed.

7. The description of the central core in the filaments seems to be missing from the revised text (was line 142-183 in the previous version). The central core is clearly visible in images 1 c, d, e. The explanation in the rebuttal raises more questions. If the thick outer layer is debris, then the actual diameter of the living cell is the diameter of the central mold. The central mold is thin or very thin, not the diameter of filaments of type 'A'.

Response: It seems that the reviewer is referring to two different structures. A few rare filaments (fig. 1c of the original submission to Nature; see image to right) do have an irregular outer layer that we interpreted in the revised manuscript as debris attached on the filaments. For these filaments, even the debris layer is excluded from the measurement, the filament diameter still falls in the range of type “A” filaments (5–10 μ m in diameter).

The reviewer also refers to filaments illustrated in fig. 1c–e of the revised manuscript (i.e., the original submission to Nature Communications; see image to right). These filaments have a very thin and discontinuous central strand, which has been described in the revised text (Line 96–98, 212–217). While the origin of the central strand remains unclear, in the revised text we infer that it is a degradational structure and represents the shrunken or condensed cytoplasm.

Revised text (Line 96–98):

“An axial strand up to half of the full filament diameter may be present in some

translucent Type A filaments (Fig. 2c–e).”

Revised text (Line 212–217):

“It is unclear that whether the axial strand in some of the translucent Type A filaments is a bona fide biotic structure, although similar structures have been reported from various fungus-like microfossils⁵⁰⁻⁵². Considering that an axial strand is absent in most Doushantuo filaments or in unambiguous fungal fossils^{7,10,14}, it more likely represents a degradational structure related to the shrinkage of the cellular cytoplasm⁵².”

8. The thin central strands in images 1 c, d, e, and also in fossils from Ivarsson, are not found in bona fide fossils of fungi.

Response: Agree. As the central strand is probably a degradational structure, it does not occur in all specimens (i.e., a central strand would not be present in filaments where the cytoplasm is completely degraded).

9. Could the filaments be interpreted as sheaths? What is known about deposition of silica or carbonates in sheaths or around filaments?

Response: See response to comment #4, where we present evidence (i.e., cell fusion) inconsistent with a sheath interpretation.

There have been many studies documenting the deposition of silica and carbonate minerals in sheaths and around filaments (Jones et al., 1997; Kolo et al., 2007). While it is possible that sheaths can be fossilized, we do not favor a sheath interpretation for the Doushantuo filaments, which show evidence of cell fusion, because a thick sheath would impede cell fusion. Instead, we prefer the interpretation that the Doushantuo fossils are fossilized filaments without a thick sheath.

10. If only the central strands in Fig. 1c etc are from the organism, what's the evidence that single, outer layer is not also debris, if it is similar in diameter and morphology to 'debris' shown in Fig. 1c?

Response: See response to comment #7. The central strand is discontinuous and much thinner than other filaments. We tentatively interpret the central strand as a degradational structure. In our collection, only a few specimens have a debris layer. Most filaments do not have a debris layer.

11. Is it possible to assess cytoplasmic anastomosis if only outer debris can be observed?

Response: See response to comment #7. The few specimens that do have a debris layer are not anastomosed. Specimens that show H- and ladder-like structures and anastomosing networks do not have a debris layer.

Other minor points

12. Line 39. The other refs are fine, but 'the Middle Ordovician Guttenberg Formation in Wisconsin¹⁷' probably represents contamination (see book, Fossil Fungi, 2015 by Taylor et al). Evidence of synchronicity is not presented and the cracked dolomite outcrop likely allowed penetration of more recent roots.

Response: This reference (Redecker et al., 2000) has been removed in the revised text.

13. Line 100. 1a nor 3c curve but they are not 'superficially resembling clamp connections of the Basidiomycota' unless any curving biological side branch superficially resembles a clamp. In 1a, the curved branch is growing past the maternal filament, unlike a clamp tip that would fuse with the maternal filament. In 3c, the filament is growing away from the maternal hypha before hooking. The lack of septa rules out clamps. Consider deleting, or add an explanation about why, as the authors correctly recognized, the hook can't be considered part of a clamp.

Response: The phrase “superficially resembling clamp connections of the Basidiomycota” has been deleted in the revised text (Line 103–106).

14. lines 120-121. Clarify what it is about the EDS elemental maps that indicate organic

matter, not just pyrite. FTIR and Raman are, however, convincing.

Response: EDS elemental maps cannot detect the trace amount of organic matter from the fossils. We have revised the text to avoid confusion (Line 123–126).

Revised text (Line 123–126):

“They are three-dimensionally pyritized, as revealed by EDS elemental maps of FIB-manufactured and petrographic sections (Supplementary Figs. 5; 6), but some contain a trace amount of organic matter, as documented by FTIR data (Fig. 6) and Raman spectroscopic data (Fig. 7).”

15. Line 160. Wording/cladistics. No extant organisms, zygomycetes included, are 'lower' in a phylogenetic sense. It's necessary to consider, not the characters of modern zygomycetes, but rather traits in the common ancestor of zygomycetes and their sister clades.

Response: We have replaced “lower fungi” with “non-Dikarya fungi” (Line 189–191) and replaced “higher fungi” with “Dikarya” in the revised text (Line 191–195).

16. lines 178-180. Almost all fungi, and possibly also some bacteria produce intercalary 'chlamydo spores.' The significance of these structures isn't easy to interpret. Although the name suggests that they are spores, whether they have spore-like function is unclear.

Response: Thanks for this information. The text has been revised accordingly (Line 207–209).

Revised text (Line 207–209):

“Similar to the Doushantuo microfossils, modern zygomycetes, as well as many other fungi, produce both intercalary and terminal (sometimes chained) chlamydo spores that are co-axially aligned with and attached to filamentous hyphae⁴⁹.”

17. Line 180. No basis for 'symbiosis'. For 'symbiosis', it would, at the least, be necessary to demonstrate that both organisms had been alive at the same time. The filaments could be saprotrophic.

Response: If the filaments were saprotrophic and the large spheres were necrotic biomass, then the filaments should be better preserved (or less degraded) than the large spheres. However, microscopic observations and microCT data show that the large spheres were preserved three-dimensionally, without evidence for strong degradation, shrinkage, or collapse. Thus, we tentatively infer that the filaments, small spheres and large spheres were alive and fossilized at the same time. The main conclusion of this paper, however, does not depend on the interpretation of the larger spheres as symbiotic autotrophs.

18. Line 184–209. Clarify how microbes could have entered the sheet cracks. Supplementary Fig. 1 suggests that the cracks were deep below ground. The added text explains more about possible paths to fossilization, but does not discuss how microbes found their way into caves.

Response: The sheet-cracks are physically connected with the karstic surfaces atop the cap dolostone, which is 1–4 m in thickness. The formation of the sheet-cracks and karstic surface requires groundwater percolation, which could have brought the microbes to the sheet-cracks. Supplementary Figure 2 (the original Supplementary Figure 1) has been revised to more precisely represent the thickness of cap dolostone. The text has also been revised accordingly.

Revised text (Line 228–233):

“As the sheet-cracks were being filled, fungal organisms, probably along with other micro-organisms (considering that the large spheres may represent symbiotic organisms living together with the fungi), colonized the cryptic sheet-crack cavities that were physically connected with a karstic surface atop of the ca. 1–4 m thick cap dolostone (Supplementary Figure 1e).”

19. Extended Fig. 6. Should present more available data from fossils, including cyanobacteria and other prokaryotes. The Doushantuo filament ratios are significantly different from all extant organisms and this suggests that R3/R2 changes with diagenesis.

Response: More FTIR data of algal and prokaryotic fossils have been added to the revised Figure 6 (the original Extended Data Figure 6). We agree that R3/R2 could be altered by diagenesis. This may explain the difference in R3/R2 ratios between extant and fossil prokaryotes.

20. Extended Fig. 8. Add more explanation to the legend to aid in interpretation. What does it show about the nature of the remains of the fossil? Does it just show that the filaments are pyritized? Specify the location of the fossil, relative to the matrix in images e through p?

Response: The legend of Supplementary Figure 5 (the original Extended Data Figure 8) has been revised accordingly to show the location and nature of the fossil and matrix (Line 340–351 in Supplementary Information).

Revised text (Line 340–351 in Supplementary Information):

“Supplementary Figure 5 | TLM, FIB-SEM, and STEM of Type A filament. a, TLM of Type A filament (opaque) in silica matrix (translucent), with white line denoting position of FIB cut. **b,** Back scattered electron SEM micrograph of specimen in **a**. **c,** Secondary electron (SE) SEM micrograph of left part of specimen in **b** before FIB cut. **d,** SE SEM micrograph after initial FIB step-cut. **e,** SE SEM micrograph after FIB step-cut and thinning. **f,** STEM overview of ultrathin foil corresponding to rectangle in **e**. **g,** Magnification of rectangle in **f** (with a 90° clockwise rotation). The bright field represents the filament and the dark field represents the matrix in **e–g**. **h–l,** EDS elemental maps of **g**, with elements marked in upper right. The filament is enriched in iron and sulfur whereas the matrix is enriched in silicon and oxygen, indicating that the filament is pyritized. Analyzed specimen is from Datang (sample

16DT-2).”

21. Extended Fig. 9. Again, add interpretation to legend.

Response: The legend of Supplementary Figure 6 (the original Extended Data Figure 9) has been revised accordingly to show the location and nature of the fossil and matrix (Line 354–370 in Supplementary Information). EDS maps of a large sphere has been added in this figure.

Revised text (Line 354–370 in Supplementary Information):

“Supplementary Figure 6 | TLM, SEM, FIB-SEM, STEM, and elemental maps of a large sphere (a–h) and a small sphere (i–l). a, TLM of a large sphere in cross section (opaque circular rim). The sphere is filled and surrounded by translucent silica, and it is penetrated by several Type A filaments (opaque elongate structures). White line denotes position of FIB cut. **b**, Secondary electron (SE) SEM micrograph of specimen in **a** after initial FIB step-cut. The bright field represents the pyritized sphere and filaments in cross section, whereas the dark field represents the silica matrix. **c**, SE SEM micrograph after FIB step-cut and thinning. **d**, STEM micrograph of ultrathin foil corresponding to rectangle in **c** (with a 90° clockwise rotation). **e–h**, EDS elemental maps of **g**, with elements marked in upper right. The wall of the large sphere is enriched in iron and sulfur and the matrix enriched in silicon and oxygen, indicating that the wall of the large sphere is pyritized. **i**. Back scattered electron SEM micrograph of a small sphere in cross section. The bright ring represents the wall of the small sphere and the surrounding dark field represents the silica matrix. **j–l**, EDS elemental maps of **i**, with elements marked in upper right. The wall of the small sphere is enriched in iron and sulfur and weakly elevated in phosphorus, indicating that it is pyritized. Specimens are from Datang (**a–h** from sample 17DT-A-9a; **i–l** from sample 16DT-2).”

In summary, the fossils are interesting. The authors have presented good arguments for excluding modern algal groups but the edits to strengthen the justification for a fungal affinity are not convincing. The problems with a fungal interpretation should be

confronted.

Response: We appreciate the reviewer's many constructive comments that helped us to think more deeply about these fossils. As suggested by the Editor, we have provided a more balanced discussion about the affinity of the Doushantuo microfossils in the revised text (Line 148–217). Alternative interpretations, including sulfur-oxidizing bacteria, stigonematalean cyanobacteria, eukaryotic algae, and particularly actinobacteria, have been discussed in greater detail. Evidence for and against an actinobacterial interpretation has been detailed in a full paragraph in the revised text (Line 176–188).

Revised text (Line 148–217):

“Several groups of fossil and extant micro-organisms provide potential interpretive analogs for the Doushantuo filaments described here. Pyritized filamentous fossils from the ~3,235 Ma Sulphur Springs Group²⁵ and the ~1,800 Ma Duck Creek Formation²⁶ in Australia, the latter interpreted as sulfur bacteria, are similar to the Doushantuo filaments in terms of filament size and preservation, but these fossil filaments do not branch or fuse. Some extant sulfur-oxidizing bacteria have a filamentous construction (for example, *Beggiatoa* and *Thioploca*²⁷), but they do not develop any branches either²⁷.

Stigonematalean cyanobacteria (for example, *Fischerella*²⁸ and *Mastigocladopsis*²⁹) can develop true branching filaments with differentiated cells, i.e., akinetes and heterocysts, surrounded by tubular sheaths^{30,31}. However, no stigonematalean cyanobacteria are known to develop aseptate trichome (Komárek, 2013). The extracellular sheaths of filamentous cyanobacteria are more resistant to degradation than cellular trichomes and thus have greater preservation potential³². When only the sheaths but not the cellular trichomes are preserved, stigonemataleans can conceivably look like the aseptate Doushantuo filaments. However, to the best of our knowledge, neither the sheaths nor the trichomes of stigonematalean filaments can fuse to form H- or ladder-like branches or networks.

Many eukaryotic algae are also characterized by a true branching organization. However, only a handful of algal groups are characterized by siphonous or

siphonocladous filaments that superficially resemble aseptate filaments³³⁻³⁵. For example, extant siphonocladaleans (e.g., *Cladophoropsis*³⁶ and *Rhizoclonium*³⁷), vaucheriaceans (e.g., *Vaucheria*³⁸), and rhodophyceans (e.g., *Griffithsia*³⁹) develop siphonous or siphonocladous thalli^{33,34}, but again their filaments do not fuse. Some multicellular zygnemataceans (a group of freshwater green algae, for example, *Spirogyra*⁴⁰) can develop H- or ladder-like structures during conjugation (a sexual reproduction process)³³, but their filaments are septate and do not form mycelium-like networks.

Actinobacteria and fungi, both of which can form mycelial networks of branching filaments, are better extant analogs for the Doushantuo filaments than those mentioned above. Actinobacteria can have repeatedly branching filaments that form radial mycelia^{41,42}, resembling the Doushantuo filaments. Some actinobacteria can have aseptate filaments, and others can produce spores^{41,42} that are morphologically comparable to small spheres described here. However, the diameters of these actinobacterial filaments and spores (~0.15–1.5 μm and ~1 μm in diameter, respectively) are usually smaller than those of the Doushantuo microfossils^{9,42}. Importantly, unlike the Doushantuo filaments, actinobacteria characteristically do not form filamentous anastomosis of network⁴³. There are rare reports of anastomoses in several strains of the actinobacterial genus *Streptomyces* in the 1950s and 1960s^{44,45}, but these anastomosal structures remains unconfirmed or they could be derived features within the actinobacteria given their restricted occurrence in the genus *Streptomyces*^{44,45}.

A better interpretive analog for the Doushantuo filaments is modern fungi, particularly non-Dikarya fungi such as zygomycetes (a paraphyletic group including the Mucoromycota and Zoopagomycota)^{46,47}. Unlike the Dikarya (a monophyletic group consisting of Basidiomycota and Ascomycota) that only produces septate filamentous hyphae, zygomycete hyphae are mostly aseptate and can branch monopodially and dichotomously, similar to the Doushantuo filaments (Figs. 2a–h; 3a–e; 4b, g–i, k; Supplementary Fig. 3a–f, j). Cell fusion is a common feature among modern fungi, where filamentous hyphae can fuse to form H- and ladder-like branches (e.g., in *Neurospora*⁴⁸ of the Ascomycota) or interconnected mycelial network (e.g., in nematode-trapping fungi in the Ascomycota such as *Arthrobotrys* and *Dactylella*¹²). Cell fusion, anastomosing hyphae, and mycelial networks also occur in many fungi of

the Mucoromycota (e.g., Mortierellomycotina⁴⁷). In addition, sexual reproduction in most zygomycetes also involves cell fusion, the formation of an H-configuration, and the eventual production of zygospores⁴⁹. Thus, when all evidence is considered, the H- and ladder-like branching systems, as well as the filamentous networks, of the Doushantuo microfossils are best compared with fungal analogs.

In light of a possible fungal affinity of the Doushantuo filaments, the smaller spheres associated with the filamentous hyphae can be interpreted as fungal spores. Similar to the Doushantuo microfossils, modern zygomycetes, as well as many other fungi, produce both intercalary and terminal (sometimes chained) chlamydospores that are co-axially aligned with and attached to filamentous hyphae⁴⁹. We note that the large spheres are in tangential contact with and sometimes penetrated by Doushantuo fungal hyphae; it is possible that these were symbiotic organisms living together with filamentous fungi, analogous to modern ecto- and endomycorrhizal fungi. It is unclear that whether the axial strand in some of the translucent Type A filaments is a bona fide biotic structure, although similar structures have been reported from various fungus-like microfossils⁵⁰⁻⁵². Considering that an axial strand is absent in most Doushantuo filaments or in unambiguous fungal fossils^{7,10,14}, it more likely represents a degradational structure related to the shrinkage of the cellular cytoplasm⁵².”

Reviewer #2 (Remarks to the Author):

The manuscript by Pang and colleagues reports fungal microfossils preserved in sheet-cracks from the Ediacaran Doushantuo Formation in China. This is a work of importance for the understanding of late Precambrian biosphere as well as for the processes that might have led to terrestrialisation of eukaryotes. I recommend this work for publication. During my previous review of the manuscript I raised some concerns with the comparisons for biological affinities, the taxonomical nomenclature and the origin of these fungal organisms. The authors have certainly addressed all of these concerns in the present submission.

Reviewer #3 (Remarks to the Author):

The authors have in a satisfactory way responded to the suggestions from the reviewers.

In my opinion, the article is now almost fit for publication. The only thing I'm missing is information about the composition of the large spheres, given that their relationship to the filaments and the small spheres is not wholly clear. A complement to the EDS maps of a small sphere (Extended Data Figure 9) to included corresponding data for large spheres would be informative.

Response: EDS maps as well as TLM, SEM, and STEM micrographs of a large sphere have been added to Supplementary Figure 6 (Supplementary Fig. 6a–h; the original Extended Data Figure 9). The preservation of the large spheres is similar to that of the filaments and small spheres, with their walls composed of pyrite whereas the interior and exterior of large sphere are filled with quartz.

One minor point: In the abstract (line 26) the interpretation of the fossils as "eukaryotes and probable fungi" is preceded (line 23) by a more definitive "fungal microfossils". If line 23 is revised to "fungal-like microfossils" the text will be more consistent.

Response: "Fungal microfossils" has been modified to "fungus-like microfossils" as suggested (Line 22–24).

References:

- Behrendt, L., Trampe, E.L., Nord, N.B., Nguyen, J., Kühl, M., Lonco, D., Nyarko, A., Dhinojwala, A., Hershey, O.S., Barton, H., 2020. Life in the dark: far-red absorbing cyanobacteria extend photic zones deep into terrestrial caves. *Environmental Microbiology* 22, 952–963.
- Berbee, M.L., Strullu-Derrien, C., Delaux, P.-M., Strother, P.K., Kenrick, P., Selosse, M.-A., Taylor, J.W., 2020. Genomic and fossil windows into the secret lives of the most ancient fungi. *Nature Reviews Microbiology*, in press, doi: 10.1038/s41579-41020-40426-41578.
- Chang, Y., Wang, S., Chang, Y., Sekimoto, S., Clum, A., Aerts, A.L., Salamov, A.A., Yee Ngan, C., Choi, C., Lindquist, E.A., Grigoriev, I.V., LaButti, K.M., Ohm, R.A., Spatafora, J.W., 2015. Phylogenomic analyses indicate that early fungi evolved digesting cell walls of algal ancestors of land plants. *Genome Biology and Evolution* 7, 1590–1601.
- Cortona, A.D., Jackson, C.J., Bucchini, F., Van Bel, M., D'hondt, S., Škaloud, P., Delwiche, C.F., Knoll, A.H., Raven, J.A., Verbruggen, H., Vandepoele, K., De Clerck, O., Leliaert, F., 2019. Neoproterozoic origin and multiple transitions to macroscopic growth in green seaweeds. *Proceedings of the National Academy of Sciences of the United States of America* 117, 2551–2559.
- Czerwik-Marcinkowska, J., Wojciechowska, A., Massalski, A., 2015. Biodiversity of limestone caves: aggregations of aerophytic algae and cyanobacteria in relation to site factors. *Polish Journal of Ecology* 63, 481–499.
- Goodfellow, M., Kämpfer, P., Busse, H.-J., Trujillo, M.E., Suzuki, K.-i., Ludwig, W., Whitman, W.B., 2012. *Bergey's Manual of Systematic Bacteriology: Volume Five The Actinobacteria, Part A and B*, 2nd ed. Springer-Verlag, New York, 2083 pp.
- Gregory, K.F., 1956. Hyphal anastomosis and cytological aspects of *Streptomyces scabies*. *Canadian Journal of Microbiology* 2, 649–655.

- Higgins, M.L., Silvey, J.K.G., 1966. Slide culture observations of two freshwater Actinomycetes. *Transactions of the American Microscopical Society* 85, 390–398.
- Jones, B., Renaut, R.W., Rosen, M.R., 1997. Biogenicity of silica precipitation around geysers and hot-spring vents, North Island, New Zealand. *Journal of Sedimentary Research* 67, 88–104.
- Kolo, K., Keppens, E., Pr at, A., Claeys, P., 2007. Experimental observations on fungal diagenesis of carbonate substrates. *Journal of Geophysical Research: Biogeosciences* 112, G01007.
- Kom rek, J., 2013. Cyanoprokaryota: 3. Teil / Part 3: Heterocytous Genera. Springer Spektrum, Heidelberg, 1130 pp.
- Kretzschmar, M., 1982. Fossile pilze in eisen-stromatolithen von warstein (rheinisches schiefergebirge). *Facies* 7, 237–259.
- Morris, J.L., Puttick, M.N., Clark, J.W., Edwards, D., Kenrick, P., Pressel, S., Wellman, C.H., Yang, Z., Schneider, H., Donoghue, P.C.J., 2018. The timescale of early land plant evolution. *Proceedings of the National Academy of Sciences* 115, E2274–E2283.
- Nieves-Rivera,  .M., Santos-Flores, C.J., Dugan, F.M., Miller, T.E., 2009. Guanophilic fungi in three caves of southwestern Puerto Rico. *International Journal of Speleology* 38, 61–70.
- Northup, D., Carr, D., Crocker, M., Cunningham, K., Hawkins, L., Leonard, P., Welbourn, C., 1994. Biological investigations in Lechuguilla Cave. *NSS Bulletin* 56, 54–63.
- Nov kov , A., 2009. Microscopic fungi isolated from the Domica Cave system (Slovak Karst National Park, Slovakia). A review. *International Journal of Speleology* 38, 71–82.
- Popovi , S., Simi , G.S., Stupar, M., Unkovi , N., Predojevi , D., Jovanovi , J., Grbi , M.L., 2015. Cyanobacteria, algae and microfungi present in biofilm from Bozana Cave (Serbia). *International Journal of Speleology* 44, 141–149.
- Redecker, D., Kodner, R., Graham, L.E., 2000. Glomalean fungi from the Ordovician. *Science* 289, 1920–1921.
- Tang, Q., Pang, K., Yuan, X., Xiao, S., 2020. A one-billion-year-old multicellular chlorophyte. *Nature Ecology & Evolution* 4, 543–549.

Reviewers' Comments:

Reviewer #3:

Remarks to the Author:

The authors have done a thorough job of revising the manuscript, addressing the reviewers' remarks. One small glitch is that the caveats implied in the interpretations ("fungus-like" rather than "fungi", etc.) have not been consistently expressed. The concluding paragraph, lines 249-264, contains wording suggesting definitive rather than tentative identifications, e.g. "unambiguous terrestrial fungi", "these fungi", "the Doushantuo fungi". This can easily be cleaned up, and the manuscript should now be fit for publication.

~~Reviewer comments in black fonts.~~

~~Point-by-point responses to reviewers' comments and quotes from revised manuscript are in red fonts, with line numbers (Line xxx) referring to the revised track change manuscript.~~

~~Dear Dr Pang,~~

~~Your manuscript entitled "Cryptic terrestrial fungi in the early Ediacaran Period" has now been seen again by our referees, whose comments appear below. In light of their advice I am delighted to say that we are happy, in principle, to publish a suitably revised version in Nature Communications under the open access CC BY license (Creative Commons Attribution 4.0 International License).~~

~~We therefore invite you to revise your paper one last time to address the remaining concerns of our reviewers and our editorial requests in the attached documents. At the same time we ask that you edit your manuscript to comply with our policies and formatting requirements and to maximise the accessibility and therefore the impact of your work.~~

~~Response: All reviewer and editorial comments have been addressed in the revised manuscript. Specific guidelines and journal styles are followed in the revised manuscript.~~

~~Please see the attached documents, listing a number of points that must be addressed. Failure to comply with our editorial requests will cause delays in accepting your manuscript. Please also see the Nature Communications formatting instructions for further information.~~

~~SUBMISSION INFORMATION~~

~~In order to accept your paper, we require the following:~~

~~–A revised author checklist describing your response to our editorial requests (attached).~~

- ~~–A separate point by point response to the reviewers' comments, reproduced verbatim.~~
- ~~–The final version of your manuscript as a Word or LaTeX file, with all changes highlighted in the text and any tables prepared using the table menu in Word or the table environment in LaTeX.~~
- ~~–If using LaTeX, please use numerical references only for citations, and include the references within the manuscript file itself. If you wish to use BibTeX, please copy the reference list from the .bbl file, paste it into the main manuscript .tex file, and delete the associated \bibliography and \bibliographystyle commands.~~
- ~~–The complete author list provided in the manuscript file, which must match that given on our manuscript tracking system. The author list in the main manuscript file will be used during typesetting of your article.~~
- ~~–Production quality versions of each figure as a separate file containing all panels. To ensure the swift processing of your paper, please provide the highest quality versions of your images and when combining different figure parts into one file for layout, use a vector based application such as Adobe Illustrator or Microsoft Powerpoint. We recommend .ai, .eps, .pdf, .ppt. Figures divided into panels should be labelled with a lower case, boldface 'a', 'b', etc. in the top left hand corner. If resolution is not of sufficient quality, production of your paper will be held whilst replacement files are obtained. For detailed guidance on figure preparation, see <https://www.nature.com/documents/aj-artworkguidelines.pdf>~~
- ~~–Please note that we do not modify the text in figures to conform to style during the production process. Please ensure that your figures are presented accurately and adhere to the guidance provided.~~
- ~~–Any updated checklists that verify compliance with our research ethics and data reporting standards in PDF format.~~
- ~~–The final version of the Supplementary Information in one PDF file.~~
- ~~–Any Supplementary Movie, Audio, Data and Software submitted as separate files. Supplementary Data and Source Data must be provided as .xls, .xlsx or .zip files, while Supplementary Software must be supplied as .zip files.~~

~~** Please note that we do not edit Supplementary Information files; they must be finalised prior to acceptance of the paper. **~~

~~–If you wish, an interesting image (but not an illustration or schematic) for consideration as a Featured Image on the Nature Communications homepage. The file should be 1400x400 pixels in RGB format and should be uploaded as a Related Manuscript File. In addition to our home page, we may also use this image (with credit) in other journal-specific promotional material.~~

~~–Completed and signed copies of our Multimedia License to Publish (LTP) for any Featured Image suggestions (please use one form for each image and give a scientific description of the image in the 'title' field; do not use "Featured Image" as a title):
<http://www.nature.com/documents/sn1-multimedia-ltp.docx>~~

~~OPEN ACCESS~~

~~Nature Communications is a fully open access journal. Articles are made freely accessible on publication under a CC BY license (Creative Commons Attribution 4.0 International License). This license allows maximum dissemination and re-use of open access materials and is preferred by many research funding bodies.~~

~~For further information about article processing charges, open access funding, and advice and support from Nature Research, please visit
<http://www.nature.com/ncomms/about/open-access>~~

~~At acceptance, the corresponding author will be required to complete an Open Access Licence to Publish on behalf of all authors, declare that all required third party permissions have been obtained and provide billing information in order to pay the article processing charge (APC) via credit card or invoice.~~

~~Please note that your paper cannot be sent for typesetting to our production team until we have received these pieces of information; therefore, please ensure that you have this information ready when submitting the final version of your manuscript.~~

ORCID

Nature Communications is committed to improving transparency in authorship. As part of our efforts in this direction, we are now requesting that all authors identified as ‘corresponding author’ create and link their Open Researcher and Contributor Identifier (ORCID) with their account on the Manuscript Tracking System (MTS) prior to acceptance. ORCID helps the scientific community achieve unambiguous attribution of all scholarly contributions. For more information please visit

<http://www.springernature.com/orcid>

For all corresponding authors listed on the manuscript, please follow the instructions in the link below to link your ORCID to your account on our MTS before submitting the final version of the manuscript. If you do not yet have an ORCID you will be able to create one in minutes.

IMPORTANT: All authors identified as ‘corresponding author’ on the manuscript must follow these instructions. Non-corresponding authors do not have to link their ORCIDs but are encouraged to do so. Please note that it will not be possible to add/modify ORCIDs at proof. Thus, if they wish to have their ORCID added to the paper they must also follow the above procedure prior to acceptance.

To support ORCID's aims, we only allow a single ORCID identifier to be attached to one account. If you have any issues attaching an ORCID identifier to your MTS account, please contact the Platform Support Helpdesk.

POLICIES

If you opted into the journal hosting details of a preprint version of your manuscript via a link on our dedicated website

(<https://nature-research-under-consideration.nature.com>), it will remain on this site while you are revising your manuscript. If you wish to remove these details, please

email naturecommunications@nature.com indicating your manuscript number and the link on our website that was previously sent to you. For more information, please

refer to our FAQ page at

<https://nature-research-under-consideration.nature.com/posts/19641-frequently-asked-questions>

In recognition of the time and expertise our reviewers provide to Nature Communications's editorial process, as of November 2018, we formally acknowledge their contribution to the external peer review of articles published in the journal. All peer-reviewed content will carry an anonymous statement of peer reviewer acknowledgement, and for those reviewers who give their consent, we will publish their names alongside the published article. For more information, please refer to our FAQ page at <https://www.nature.com/documents/ncomms-reviewer-information.pdf>

Nature Research journals encourage authors to share their step-by-step experimental protocols on a protocol sharing platform of their choice. Where such protocols are available, please provide a DOI or other citation details in the paper. Nature Research's Protocol Exchange is a free-to-use and open resource for protocols; protocols deposited in Protocol Exchange are citable and can be linked from the published article. More details can found at <https://www.nature.com/protocolexchange/about>

We hope to hear from you within two weeks and look forward to seeing your revised manuscript; please let us know if the process may take longer.

Best regards,

Emily

Emily Jones, PhD

Senior Editor, Nature Communications

<http://orcid.org/0000-0002-0605-884X>

<http://www.nature.com/ncomms>

REVIEWERS' COMMENTS

Reviewer #3 (Remarks to the Author):

The authors have done a thorough job of revising the manuscript, addressing the reviewers' remarks. One small glitch is that the caveats implied in the interpretations ("fungus-like" rather than "fungi", etc.) have not been consistently expressed. The concluding paragraph, lines 249-264, contains wording suggesting definitive rather than tentative identifications, e.g. "unambiguous terrestrial fungi", "these fungi", "the Doushantuo fungi". This can easily be cleaned up, and the manuscript should now be fit for publication.

Response: The title, discussion section, and concluding paragraph in the main text, as well as discussion in the Supplementary Information, have been revised to tone down the definitive claim. For example, 'fungi' is replaced with 'fungus-like fossils' or 'fungus-like micro-organisms' (Line 1, 57, 220, 224, 239, 241, 268, 277). We have also made minor changes in the text, Figure 6, and Supplementary Figure 4 to improve clarity.